# P-GenRM: Personalized Generative Reward Model with Test-time User-based Scaling

**Pinyi Zhang**[1,2]*, **Ting-En Lin**[2], **Yuchuan Wu**[2]†, **Jingyang Chen**[4], **Zongqi Wang**[2]*,
**Hua Yang**[3], **Ze Xu**[2], **Fei Huang**[2], **Yongbin Li**[2], **Kai Zhang**[1]†

[1] School of Computer Science and Technology, East China Normal University
[2] Tongyi Lab, Alibaba Group    [3] Alibaba Group
[4] Institute of Science and Technology for Brain-inspired intelligence, Fudan University
`51265901020@stu.ecnu.edu.cn, kzhang980@gmail.com`
`{ting-en.lte, shengxiu.wyc}@alibaba-inc.com`

## Abstract

Personalized alignment of large language models seeks to adapt responses to individual user preferences, typically via reinforcement learning. A key challenge is obtaining accurate, user-specific reward signals in open-ended scenarios. Existing personalized reward models face two persistent limitations: (1) oversimplifying diverse, scenario-specific preferences into a small, fixed set of evaluation principles, and (2) struggling with generalization to new users with limited feedback. To this end, we propose **P-GenRM**, the first **P**ersonalized **Gen**erative **R**eward **M**odel with test-time user-based scaling. P-GenRM transforms preference signals into structured evaluation chains that derive adaptive personas and scoring rubrics across various scenarios. It further clusters users into User Prototypes and introduces a dual-granularity scaling mechanism: at the individual level, it adaptively scales and aggregates each user's scoring scheme; at the prototype level, it incorporates preferences from similar users. This design mitigates noise in inferred preferences and enhances generalization to unseen users through prototype-based transfer. Empirical results show that P-GenRM achieves state-of-the-art results on widely-used personalized reward model benchmarks, with an average improvement of 2.31%, and demonstrates strong generalization on an out-of-distribution dataset. Notably, Test-time User-based scaling provides an additional 3% boost, demonstrating stronger personalized alignment with test-time scalability[1].

## 1 Introduction

Reinforcement learning from human feedback has become a prevailing paradigm for aligning large language models (LLMs) with broadly accepted human values, such as helpfulness and harmlessness (Askell et al., 2021; Bai et al., 2022). Central to this approach is the reward model, which provides reliable scoring signals to steer the LLM's outputs towards desired behaviors (Wang et al., 2024).

While conventional alignment targets universal values, personalized alignment aims to tailor model behavior to the diverse preferences of individual users (Jang et al., 2023; Salemi et al., 2023; Zollo et al., 2024; Ryan et al., 2025). This shift poses challenges, especially in open-ended tasks like dialogue (Salemi & Zamani, 2025), where evaluation depends heavily on subjective standards: Explicit preference signals (e.g., "I prefer concise answers") are often sparse, while implicit signals (e.g., conversation history) are richer but noisy (Guan et al., 2025). Hybrid approaches that combine demographics, behavior, and context have been explored (Zhang et al., 2024b; Maghakian et al., 2022; Zhao et al., 2023; Poddar et al., 2024), but two key limitations remain: (1) Static modeling of preferences. Most methods reduce diverse and dynamic user preferences to a fixed set of evaluation rules, failing to capture scenario-dependent variability, even within a user (e.g., preferring brevity while driving but expressiveness in casual settings); (2) Weak generalization to new users. Current models struggle to adapt to new users with sparse feedback, limiting effectiveness in cold-start scenarios.

---

*Work done while interning at Tongyi Lab. †Yuchuan Wu and Kai Zhang are corresponding authors.
[1] Code available at: https://github.com/Tongyi-ConvAI/Qwen-Character/tree/main/Character-GenRM

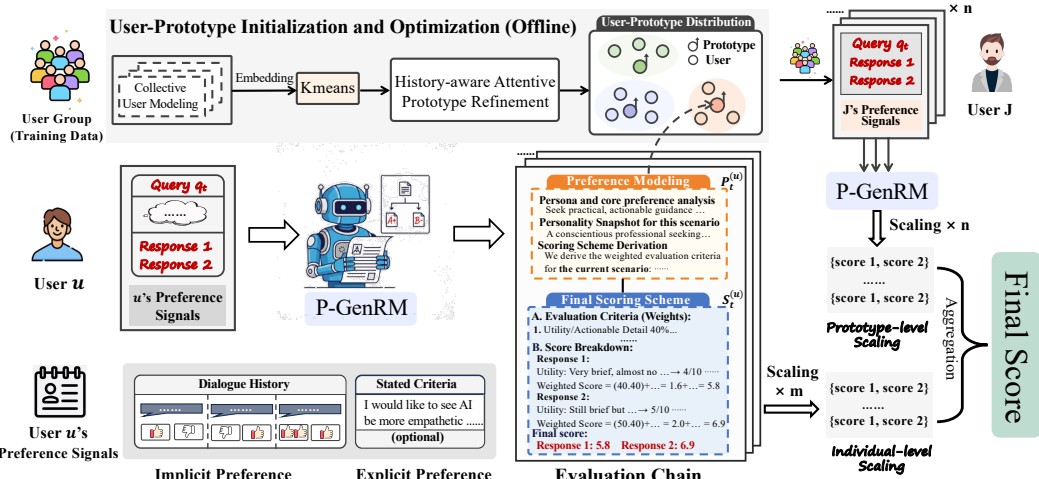

Figure 1: Workflow of P-GenRM. P-GenRM infers a scenario-specific user persona and preference analysis from hybrid preference signals, generates dynamic scoring rubrics, and assesses candidate responses accordingly. At test-time, P-GenRM can aggregate multiple individual-level scoring schemes and incorporate similar users' preferences to improve scoring accuracy and generalization.

To address these challenges, we introduce P-GenRM, a Personalized Generative Reward Model with test-time user-based scaling. Our approach leverages user personas as priors for interpreting implicit signals, enriched by explicit preference cues when available. We design a three-stage training framework: (1) Persona-guided Scoring Induction (PSI) via supervised fine-tuning, which translates hybrid preference signals into explicit evaluation chains; (2) Criteria-based Reasoning Enhancement (CRE) with reinforcement learning, which strengthens evaluation chain generation, especially in settings with missing preference information; and (3) hard-negative-aware curriculum learning, which progressively improves robustness in handling challenging instances.

After the three-stage training, the P-GenRM can now interpret multi-faceted user preference indicators into comprehensive, structured evaluation chains that derive adaptive user personas and scoring rubrics in variable contexts. We further investigate two questions: (1) *How to mitigate the noise inherent in inferred user preference*; (2) *How to develop a transfer mechanism that enhances generalization to new users by adapting learned preferences?* Inspired by the collaborative filtering algorithms (Goldberg et al., 1992; Sarwar et al., 2001) in recommendation systems, we propose a prototype-based, dual-granularity scaling mechanism to address this issue in a unified manner. As shown in Figure 1, at test-time, **P-GenRM dynamically scales the generation of scoring schemes for each user according to their individual preferences, while simultaneously incorporating ratings from similar users within the same prototype**. This mechanism reduces noise in inferred preferences and provides strong generalization to unseen users through prototype-based transfer.

Extensive experiments demonstrate that P-GenRM not only sets a new state of the art on personalized reward model benchmarks but also achieves an additional 3% gain from test-time scaling, highlighting its scalability and effectiveness in real-world personalization.

In summary, the main contributions of this paper are as follows:

1. We present P-GenRM, to the best of our knowledge, the first personalized generative reward model that transforms diverse preference signals into structured evaluation chains, including personas and rubrics, in open-domain settings.

2. We fully leverage the scalability of GenRM and propose a *Test-time User-based Scaling* mechanism, which substantially improves model performance.

3. Experimental results demonstrate that P-GenRM achieves state-of-the-art performance on personalized reward benchmarks and exhibits strong generalization to new users.

## 2 RELATED WORKS

**Personalized alignment of Large Language Models** Personalized alignment of large language models aims to tailor responses to diverse user preferences. This can be achieved both by training models with user-specific parameters (Li et al., 2023a; Wang et al., 2023b; Li et al., 2024), or steering the LLMs' behavior at inference time (Wang et al., 2023a; Salemi et al., 2023; Lee et al., 2024). We focus here on an important paradigm of the former-Personalized alignment via reinforcement learning, in which one essential component is the personalized reward model. Poddar et al. (2024) constructs a reward model conditioned on a novel, inferred user-specific latent. Chen et al. (2024) model user preferences as a convex combination of finite prototypes to construct reward models. Rame et al. (2023) promote diversity by combining a mixture of experts conditioned on user preferences to learn diverse rewards. Jang et al. (2023) decompose preferences into multiple dimensions and train separate reward models to balance different objectives. Building on user responses, Shenfeld et al. (2025) represent user-specific rewards as a linear combination of base reward functions. Ryan et al. (2025) propose SynthesizeMe, which infers synthetic personas from historical preferences to build personalized prompts and reward models, but its static design cannot adapt to context-dependent and shifting user preferences.

**User Preference Modeling** User preference modeling is a prominent research topic across diverse fields. Here, we introduce several User preference modeling methods used in LLM alignment tasks; more details can be found in Appendix A.4. Dong et al. (2023) define explicit multi-dimensional attributes to model human preferences and enhance response customizability. Lee et al. (2024) encode thousands of user-specified preferences as combinations of values within system prompts. Zhao et al. (2023) train a transformer module to predict group preferences and guide LLM generation in a few-shot setting.

**Generative Reward Models** Recently, generative reward models (GenRM) have attracted increasing attention for their strong generalization, support for test-time scaling, and effective utilization of the powerful generative capabilities of LLMs. Zhang et al. (2024a) train generative verifiers via the ubiquitous next-token prediction objective, thereby seamlessly integrating reward modeling. Li et al. (2023b) propose a generative judge to enhance generality, flexibility, and interpretability in alignment evaluation tasks. Liu et al. (2025) introduce Self-Principled Critique Tuning, which scales the generation of high-quality principles and precise critiques to enhance scalability. Zhao et al. (2025) incorporate generative chain-of-thought reasoning and code verification into process-level rewarding. Xiong et al. (2025) develop a generative judge that evaluates a policy model's intermediate reasoning steps for stepwise reward modeling.

## 3 PROBLEM FORMULATION

In a dialogue system, at turn $t$, the current user $u$ issues a query $q_t$. For each previous turn $\tau < t$, the user's preferred response and dispreferred response are denoted as $y_\tau^+$ and $y_\tau^-$. The user's historical interaction up to turn $t$ is therefore

$$H_t^{(u)} = \left\{ (q_1, y_1^+, y_1^-), \ldots, (q_\tau, y_\tau^+, y_\tau^-), \ldots, (q_{t-1}, y_{t-1}^+, y_{t-1}^-) \right\}^{(u)}. \tag{1}$$

In practice, to avoid excessive reliance on historical data, the history size is limited to $\mathbf{h}$ via random selection, i.e., $|H_t^{(u)}| = \mathbf{h}$. In addition to these implicit interaction signals, the user may also provide explicit preference criteria $E^{(u)}$ (e.g., desired style, tone, etc.). Both explicit criteria and implicit signals extracted from historical interactions are treated as signals of the user's preferences. Based on this, the personalized generative reward model is designed to first infer a context-aware textual description of the user's personalized preference modeling $P_t^{(u)}$, and subsequently to generate a set of scoring rubrics with associated weights, thereby defining a scoring process $S_t^{(u)}$ that evaluates responses based on their adherence to these rubrics. This process can be formalized as:

$$[P_t^{(u)}; S_t^{(u)}] \sim R_\theta \big( q_t, H_t^{(u)}, E^{(u)}, y_t^i \big), \quad \{s_t^i\}_{i=1}^b = \text{Extract}(S_t^{(u)}) \tag{2}$$

where $R_\theta$ denotes the personalized generative reward model parameterized by $\theta$, $s_t^i$ represents the scalar reward assigned to the $i$-th candidate response $y_t^i$, and $b$ denotes the number of candidate responses (typically $b = 2$). Extract denotes the operation of extracting each response's score from a text-based scoring process.

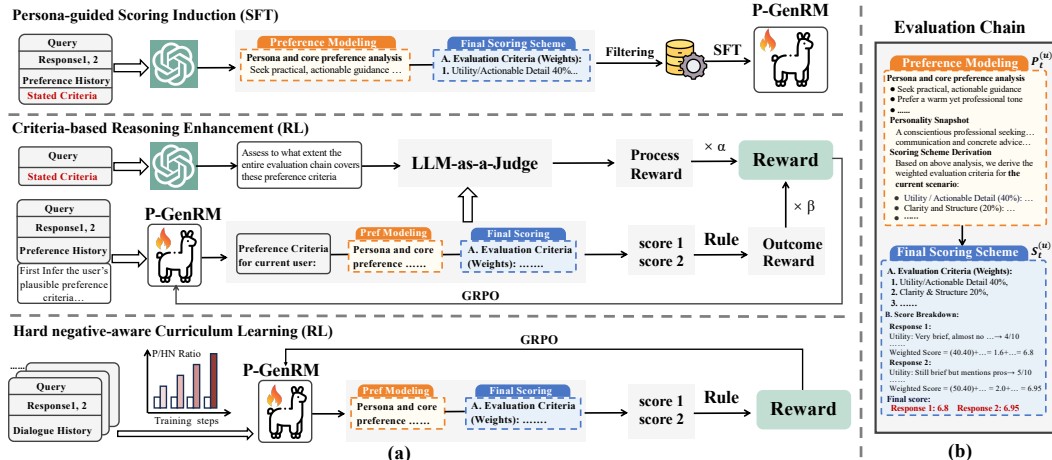

Figure 2: **(a)** The three-stage training framework of P-GenRM **(b)** An illustration of the personalized evaluation chain, showing how preference modeling and derived scoring schemes lead to interpretable, criterion-weighted judgments for responses.

## 4 METHODOLOGY

As shown in Figure 1, given the user's current query $q_t$ and preference signals $\{H_t^{(u)}, E^{(u)}\}$, P-GenRM transforms these signals into structured evaluation chains that derive adaptive personas and scenario-specific scoring rubrics, built upon our proposed three-stage training strategy:

### 4.1 MULTI-STAGE TRAINING FRAMEWORK

Our training pipeline consists of three stages, as illustrated in Figure 2: SFT equips the model with basic personalized scoring abilities; RL further improves the quality of the evaluation chains; and curriculum learning enhances the model's robustness on difficult negative cases for such highly subjective task setting.

**Persona-guided Scoring Induction (SFT).** The basic premise of P-GenRM is to integrate self-generated analyses of user preferences into the generative reward modeling process, so as to enhance scoring accuracy in subjective settings. We conduct a preliminary experiment to investigate how various preference-related factors affect the accuracy of personalized scoring. Detailed information is provided in Table 6, and the conclusions are as follows:

**(i):** User persona inferred from interaction histories can serve as an effective prior to infer the user's preference, while **(ii)** user's explicitly stated criteria further sharpen the evaluation precision.

Building on these findings, we first construct a Structured Evaluation Chain (SEC) dataset for response scoring. As shown in Figure 2, an instruct LLM is prompted with both the user's implicit and explicit preference signals $\{H_t^{(u)}, E^{(u)}\}$ to first induce the user's preference modeling $P_t^{(u)}$, consisting of a scenario-specific persona and then derive the corresponding preference criteria. Then each candidate's response $y_t^i$ is evaluated against these criteria to yield a personalized score $s_t^i$. We utilize the data filtered through rejection sampling as the SFT stage, enabling P-GenRM to acquire the initial capability of transforming hybrid preference factors into an adaptive personalized scoring scheme.

**Criteria-based Reasoning Enhancement (RL).** To generate high-quality evaluation chains in scenarios lacking explicit user feedback, we propose Criteria-based Reasoning Enhancement via reinforcement learning.

Our method builds on the standard GRPO (Shao et al., 2024) algorithm and incorporates both process-level and rule-based outcome rewards. Specifically, we instruct P-GenRM with *limited number of* history interactions to first infer plausible explicit preference from historical interactions. Based on these, P-GenRM likewise generates the evaluation chain in the Persona-guided Scoring

Induction manner. We then adopt LLM-as-a-judge (Gu et al., 2024) to score the process, where the judge is tasked to assess how well the evaluation process covers user-stated or synthetic (details in Appendix C ) explicit preferences and output a score (ranging from 0 to 1), denoted as $\mathrm{PR_t}$.

Moreover, we define rule-based outcome rewards based on the correctness of the final output $\mathrm{OR_t} = \mathbf{1}\{s_t^c > s_t^r)\}$, where $c$ and $r$ are the indices of the responses labeled as *chosen* and *rejected*, respectively. Additionally, in cases of formatting error, a penalty of $-0.1$ is imposed.

The overall reward is computed as a weighted sum of the process reward and the outcome reward.

$$\mathrm{R}_t = \alpha \cdot \mathrm{PR}_t + \beta \cdot \mathrm{OR}_t, \tag{3}$$

where $\alpha$ and $\beta$ are hyperparameters to balance the relative contributions of the two reward signals. $\mathrm{R}_t$ is further utilized to compute the advantage function in the GRPO algorithm.

$$
\begin{aligned}
J_{GRPO}(\theta) = & \mathbb{E}_{\substack{(q_t, H_t^{(u)}, y_t^i) \sim \mathcal{D} \\ \{c_t^{(k)}\}_{k=1}^K \sim \pi_{\theta_{\mathrm{old}}}}} \Bigg[ \frac{1}{K} \sum_{k=1}^{K} \frac{1}{|c_t^{(k)}|} \sum_{j=1}^{|c_t^{(k)}|} \Bigg\{ \min\Bigg( \frac{\pi_\theta(c_{t,j}^{(k)}|q_t, H_t^{(u)}, y_t^i, c_{t,<j}^{(k)})}{\pi_{\theta_{\mathrm{old}}}(c_{t,j}^{(k)}|q_t, H_t^{(u)}, y_t^i, c_{t,<j}^{(k)})} A_t^{(k)}, \\
& \mathrm{clip}\Bigg( \frac{\pi_\theta(c_{t,j}^{(k)}|q_t, H_t^{(u)}, y_t^i, c_{t,<j}^{(k)})}{\pi_{\theta_{\mathrm{old}}}(c_{t,j}^{(k)}|q_t, H_t^{(u)}, y_t^i, c_{t,<j}^{(k)})}, 1 - \varepsilon, 1 + \varepsilon \Bigg) A_t^{(k)} \Bigg) \Bigg\} - \beta D_{\mathrm{KL}}(\pi_\theta \| \pi_{\mathrm{ref}}) \Bigg]
\end{aligned}
$$

where $c_t^{(k)} = [P_t^{(u)}; S_t^{(u)}]$ denotes the $k$-th sampled structured evaluation chain, $|c_t^{(k)}|$ denotes the length of $c_t^{(k)}$, and $A_t^{(k)}$ is the relative advantage computed from the corresponding reward $R_t^{(k)}$.

This reward function encourages the P-GenRM not only to generate correct outputs but also to ensure that the evaluation chain reflects the user's preferences. Moreover, by adjusting the weighting factor $\alpha$, the model's sensitivity to explicit user preferences can be modulated. This helps prevent the model from overfitting to specific preference dimensions and preserves its ability to freely explore the broader preference space. We provide the impact of variations in $\alpha$ and $\beta$ in Appendix A.3

**Hard negative-aware Curriculum Learning (RL).** Following the SFT and RL stages, we introduce a hard negative-aware curriculum learning (Bengio et al., 2009) phase to enhance the P-GenRM's ability to learn from challenging cases. Specifically, we gradually increase the proportion of hard negatives during the training phase. In addition, to enable a larger exploration space for hard negative samples, we disable process-level reward here. Accordingly, the objective function in this stage retains the same form as in the previous stage, except that the reward function $\mathrm{R}_t$ omits the process reward $\mathrm{PR}_t$ .

## 4.2 TEST-TIME USER-BASED SCALING

In this subsection, we propose a Test-time User-based Scaling mechanism to jointly address the two key challenges of personalized reward modeling: (1) the inherent noise in inferring user preferences; (2) the limited generalization on new users with sparse feedback. It comprises two aspects: offline prototype initialization and optimization, and test-time dual-granularity scaling, as detailed below.

### 4.2.1 OFFLINE PROTOTYPE INITIALIZATION AND OPTIMIZATION

**User Prototype Initiation.** We first employ Qwen3-Embedding-0.6B(Zhang et al., 2025) to compute an embedding for each $P_t^{(u)} \in \mathbb{R}^d$. We then concatenate these embeddings to form $\mathbf{P}$, an overall cross-scenario user-preference embedding matrix. Subsequently, we apply K-means clustering to $\mathbf{P}$ to obtain $k$ centroids, denoted as $\mathbf{A} \in \mathbb{R}^{k \times d}$, which serve as the initial user prototypes.

**History-aware Attentive Prototype Refinement**

The goal of prototype refinement is to transform them from mere semantic centers into effective priors that can represent subordinate users' preference choices, which consists of the following steps

(1) *Historical selection and prior construction.* Given the j-th prototype $a_j = A_{[j,:]}$ and the historical records of its associated user $u$ at turn $t$: $H_t^{(u)} = \big\{(q_\tau, y_\tau^+, y_\tau^-) \,\big|\, \tau \in \mathrm{Random}(t-1, \mathbf{h})\big\}^{(u)}$, $\mathrm{Random}(t-1, \mathbf{h})$ denotes that we randomly sample $\mathbf{h}$ items from the history up to step $t$.

For brevity, we assume that these letters represent their embeddings, where $q_\tau \in \mathbb{R}^d$ is the $\tau$-th input embedding and $y_\tau^+$, $y_\tau^- \in \mathbb{R}^d$ are the embeddings of positive and negative feedback, each history triple is first encoded as $o_\tau = \sigma\big(W \cdot \text{concat}(q_\tau,\ y_\tau^+ - y_\tau^-)\big)$

A prototype-augmented attention mechanism assigns importance weights to historical records:

$$v_H = \sum_{\tau=1}^{\mathbf{h}} \alpha_\tau o_\tau, \quad \alpha_\tau = \text{softmax}_\tau\left(\frac{o_\tau^\top q_t}{\sqrt{d}} + \rho\frac{o_\tau^\top a_j}{\sqrt{d}}\right) \tag{4}$$

This procedure ensures that the prototype selectively exploits those historical records most informative for the current query.

(2) *Discriminative prior update with regularization.* We integrate the salient historical information with the prototype representation, forming a prior that is expected to guide the current query $q_t$ in effectively discriminating between $y_\tau^+$ and $y_\tau^-$.

$$z_t = a_j + \lambda_q W_q q_t + \lambda_s W_s v_H, \quad z \in \mathbb{R}^d \tag{5}$$

The prior $z_t$ is then employed to quantify the preference for $y_t^+$ over $y_t^-$. This preference is captured by a discriminative score difference, $\Delta_t$, which is then used to formulate a pairwise loss $\mathcal{L}_{\text{pair}}$. The model aims to maximize this difference by minimizing the loss:

$$\Delta_t = z_t^\top y_t^+ - z_t^\top y_t^-, \qquad \mathcal{L}_{\text{pair}} = -\log\sigma(\Delta_t) \tag{6}$$

To avoid excessive drift, the overall objective function for a given prototype $a_j$ is augmented with two regularization terms. The final loss is defined as:

$$\mathcal{L} = \mathcal{L}_{\text{pair}} + \lambda_{\text{cent}}\|a_j - \mu_j\|_2^2 + \lambda_{\text{tr}}\|a_j - p_j\|_2^2. \tag{7}$$

Here, $\mu_j$ is the center of the $j$-th cluster (i.e., the mean of its associated sample embeddings), and $p_j$ is the state of the prototype $a_j$ from the previous update step. The first regularizer encourages the prototype to stay close to its cluster center, while the second ensures that the prototype's evolution remains smooth across updates. The hyperparameters $\lambda_{\text{cent}}$ and $\lambda_{\text{tr}}$ control the strength of these regularization effects. This $\mathcal{L}$ will backpropagate to update the prototype $a_j$. We perform this procedure on all samples within each prototype, subsequently reassigning them to the nearest prototype. An algorithmic procedure is in Appendix 1

### 4.2.2 TEST-TIME DUAL-GRANULARITY SCALING

We propose two scaling mechanisms that fully leverage the inherent test-time scalability of GenRM: **individual-level scaling** and **prototype-level scaling**. The former performs parallel sampling on the user's current query to generate multiple scoring schemes, while the latter incorporates preference signals from similar users when evaluating the candidate responses. Details are as follows:

We employed the resulting updated user–prototype distribution in 4.2.1 to perform test time scaling. Specifically, given the user's query $q_t$, preference signals $H_t^u$, candidate response $y_t^i$, P-GenRM performs parallel sampling to constructs $m$ individual-level scoring schemes that reflect the user's own preference. Moreover, given the user's preference embedding, P-GenRM assign it to the nearest prototype and then select the $n$ most similar users $\{u_w\}_{w=1}^n$ according to the embedding. Based on these references, P-GenRM simultaneously incorporates $n$ additional scores inferred from the preferences of similar users. This process can be formalized as follows:

$$S_{t,x}^i \sim R_\theta\big(q_t, H_t^{(u)}, y_t^i, P_{t,x}^{(u)}\big), \quad \big(S_t^i\big)^{(u_w)} \sim R_\theta\big(q_t, H_t^{(u_w)}, y_t^i, P_t^{(u_w)}\big) \tag{8}$$

$$s_t^i = \frac{1}{m}\sum_{x=1}^m \text{Extract}(S_{t,x}^i) + \frac{1}{n}\sum_{w=1}^n \text{Extract}\big((S_t^i)^{(u_w)}\big) \tag{9}$$

where $P_{t,x}^{(u)}$ is the preference analysis obtained from the $x$-th sampling, $P_t^{(u_w)}$ represents the preference signals of a similar user $u_w$, Extract denotes the operation for extracting the score from the generated textual analysis, and $s_t^i$ is the final score assigned to $y_t^i$ after aggregating all scaled results.

The effectiveness of this approach can be attributed as follows: At the *individual level*, it explores multiple hypotheses about a user's preferences to obtain richer and more robust scoring rubrics, while at the *prototype level* it refines these inferences using preferences from similar users. Moreover, for new users under sparse historical data, assigning them to suitable prototypes allows the model to approximate their preferences through those of similar users.

## 5 EXPERIMENTS

### 5.1 DATASETS AND EXPERIMENTAL SETTINGS

We evaluate P-GenRM and various baselines on three popular personalized alignment datasets: **Chatbot Arena-personalized and Prism-personalized:** Ryan et al. (2025) devise a data-filtering pipeline to extract challenging and highly personalizable user data from Chatbot Arena (Zheng et al., 2023) and PRISM (Kirk et al., 2024) [2].

**LaMP-QA:** LaMP-QA(Salemi & Zamani, 2025) evaluates personalized long-form question answering by testing how well LLMs generate informative, coherent, and contextually relevant answers given a user profile. More information on these three datasets is in Appendix B.1

**Experimental Settings:** For experiments on PersonalRewardBench, we implement both P-GenRM and multiple baseline models across two model scales: LLaMA-3.1-8B and LLaMA-3.1-70B. Experiments on LaMP-QA are conducted only with LLaMA-3.1-8B. Experiments involving 8B/70B models utilize 8/32 GPUs, respectively. LLaMA-3.1-70B training is performed using LoRA (Hu et al., 2022). The weight parameters $\alpha$ and $\beta$ of process reward and outcome reward in Equation 3 are 0.5 and 1.0 in our RL stage training. The impact of variations in their values is shown in Table 7 in Appendix A.3. The model used for embedding the prototype is Qwen3-Embedding-0.6B. The instruction model used in Persona-guided Scoring Induction is OpenAI o3 (https://openai.com)

### 5.2 PERSONALIZED ALIGNMENT PERFORMANCE ON PERSONALREWARDBENCH

**Baselines.** We compare P-GenRM with various baselines on PersonalRewardBench:

*In-Context LLM as a Judge*: We provide the LLMs with varied personalization information in the prompt, instructing them to infer the user's preference and determine which response is superior.

*Finetuned Reward Models:* (a) Following Ryan et al. (2025), we train the Bradley–Terry (Bradley & Terry, 1952) Reward Model on the unfiltered data from PersonalRewardBench to capture and fit the overall preference distribution of this specific population. (b) We implement other Existing Personalized Reward Models with distinct motivations and methodologies, including GPO(Zhao et al., 2023), VPL(Poddar et al., 2024), PAL(Chen et al., 2024), and SynthesizeMe. (Ryan et al., 2025). More details of these baselines are in Appendix B.2.

**Overall Results.** Performances of different baselines and P-GenRM are shown in Table 1. We find that P-GenRM consistently outperforms the previous state-of-the-art (SOTA) across model scales. Specifically, on the 8B model, P-GenRM achieves an average improvement of 2.77% over the prior SOTA, while on the 70B model (trained with LORA), it delivers an average gain of 1.99%. Furthermore, P-GenRM-8B surpasses the previously best-performing 70B model by an average of 1.04%. In addition, as shown in Table 4, P-GenRM also significantly outperforms the leading proprietary model, OpenAI-o3, instructed with strong prompting. Moreover, to ensure that the preferences of minority groups are given equal consideration, we provide additional comparative experiments in Appendix A.5 using **macro accuracy** as the metric, where accuracy is computed separately for each persona group and then averaged across all groups. P-GenRM still achieves the highest macro accuracy (65.21%) among all evaluated baselines and does not overfit to any majority persona.
**The effectiveness of Test-time User-based Scaling.** Leveraging the test-time scalability of P-GenRM, P-GenRM generates multiple scoring schemes tailored to the current user and aggregates the corresponding results to improve scoring accuracy. More importantly, **by incorporating scoring schemes from similar users, P-GenRM achieves higher accuracy while requiring only a modest increase in scaling operations**. For instance, the best result comes from the setting of Ind-16 and Pro-8, which surpasses the performance of Ind-32 with fewer scaling steps (16+8), yielding an

---

[2]Ryan et al. (2025) name the combined benchmark **PersonalRewardBench**, which we also adopt hereafter.

Table 1: Results on PersonalRewardBench. P-GenRM outperforms all baselines on both datasets and model scales, while Test-time User-based Scaling brings further gains. Best and second-best results are marked in **bold** and underline. Ind and Pro denote the Individual and Prototype level scaling, respectively. Results are reported as "mean ± standard error" over 5 independent runs.

| Base Model | Chatbot Arena-Personalized | | PRISM-Personalized | |
|---|---|---|---|---|
| | Llama 3.1 8B | Llama 3.1 70B | Llama 3.1 8B | Llama 3.1 70B |
| **In-Context LLM as a Judge** | | | | |
| Default | $56.37 \pm 2.16\%$ | $57.02 \pm 2.06\%$ | $52.04 \pm 0.54\%$ | $54.02 \pm 0.83\%$ |
| + CoT | $57.05 \pm 2.05\%$ | $57.61 \pm 1.82\%$ | $52.58 \pm 0.85\%$ | $54.09 \pm 0.64\%$ |
| + Demographics | — | — | $52.96 \pm 0.59\%$ | $54.21 \pm 0.56\%$ |
| + Preference History | $58.53 \pm 1.45\%$ | $58.65 \pm 2.10\%$ | $56.24 \pm 0.48\%$ | $57.11 \pm 0.76\%$ |
| + SynthesizeMe | $61.07 \pm 2.01\%$ | $63.14 \pm 1.91\%$ | $54.70 \pm 0.87\%$ | $58.19 \pm 0.61\%$ |
| + Persona-guided Scoring Induction (Ours) | $62.20 \pm 1.41\%$ | $65.55 \pm 1.64\%$ | $58.33 \pm 0.51\%$ | $61.61 \pm 0.72\%$ |
| **Finetuned Reward Models** | | | | |
| **Bradley-Terry** | | | | |
| Finetuned Reward Model | $67.21 \pm 2.07\%$ | $71.12 \pm 1.71\%$ | $63.27 \pm 0.62\%$ | $63.44 \pm 0.77\%$ |
| **Existing Personalized Reward Model** | | | | |
| GPO | $57.87 \pm 2.20\%$ | $58.50 \pm 2.37\%$ | $57.29 \pm 1.06\%$ | $59.16 \pm 1.25\%$ |
| VPL | $58.12 \pm 2.64\%$ | $59.02 \pm 2.08\%$ | $58.25 \pm 0.68\%$ | $59.70 \pm 1.10\%$ |
| PAL | $57.31 \pm 2.49\%$ | $59.40 \pm 2.69\%$ | $56.74 \pm 1.18\%$ | $57.75 \pm 0.83\%$ |
| FT RM + SynthesizeMe | $69.78 \pm 1.98\%$ | $72.05 \pm 2.24\%$ | $62.84 \pm 0.85\%$ | $63.74 \pm 0.66\%$ |
| **Personalized Generative Reward Model** | | | | |
| P–GenRM | **$72.68 \pm 1.85\%$** | **$73.42 \pm 1.74\%$** | **$65.32 \pm 0.56\%$** | **$66.21 \pm 0.76\%$** |
| **Test-time User-based Scaling** | | | | |
| + Ind-8, Pro-4 | $74.30 \pm 1.60\%$ | — | $67.54 \pm 0.58\%$ | — |
| + Ind-16, Pro-8 | **$75.92 \pm 1.70\%$** | — | **$68.06 \pm 0.69\%$** | — |

average improvement of 2.99 % over P-GenRM itself and a substantial average over both SOTA open-source and proprietary models. This demonstrates the effectiveness of our proposed Test-time User-based Scaling strategy. Notably, simply increasing the number of ratings from similar users does not necessarily yield performance gains (as shown in the last two rows of Table 4), further underscoring the highly user-specific nature of the personalized scoring task.

Notably, we measured the end-to-end inference time of P-GenRM-8B at different scaling levels on Chatbot Arena-Personalized test set and compared it against several baselines. The proposed test-time user-based scaling incurs only a limited increase in inference time while yielding superior performance and retaining lower latency than prior SOTA methods. Details are presented in Appendix A.9

**Ablation Study.** We conduct ablation studies to assess the contribution of each component of P-GenRM (Table 2). Removing any component leads to a significant performance drop. Notably, the RL-stage ablation indicates that both process and outcome rewards are necessary for this task.

Table 2: Ablation studies of P-GenRM components: CL (Curriculum Learning), PR (Process Reward), OR (Outcome Reward). Results are reported as "mean ± standard error" over 5 independent runs.

| | Chatbot Arena | PRISM |
|---|---|---|
| P-GenRM | **$72.68 \pm 1.85\%$** | **$65.32 \pm 0.56\%$** |
| w/o CL | $71.07 \pm 1.44\%$ | $63.82 \pm 0.64\%$ |
| w/o CL, PR | $70.22 \pm 1.74\%$ | $62.70 \pm 0.73\%$ |
| w/o CL, OR | $69.05 \pm 1.59\%$ | $60.94 \pm 0.77\%$ |
| w/o CL, RL | $66.76 \pm 1.42\%$ | $57.08 \pm 0.89\%$ |
| w/o CL, RL, SFT | $56.37 \pm 2.16\%$ | $52.04 \pm 0.54\%$ |

Table 3: Comparison of adaptive (PSI, Persona-guided Scoring Induction) and static (SMe, SynthesizeMe) persona methods across base models. Results are reported as "mean ± standard error" over 5 independent runs.

| | Chatbot Arena | PRISM |
|---|---|---|
| Qwen3–8B | $61.82 \pm 1.47\%$ | $55.01 \pm 0.77\%$ |
| Qwen3–8B + SMe | $62.57 \pm 1.84\%$ | $56.33 \pm 0.91\%$ |
| Qwen3–8B + PSI (ours) | **$64.22 \pm 1.58\%$** | **$58.01 \pm 0.83\%$** |
| o3 | $64.47 \pm 1.62\%$ | $56.34 \pm 0.64\%$ |
| o3 + SMe | $67.73 \pm 1.94\%$ | $58.49 \pm 1.22\%$ |
| o3 + PSI (ours) | **$69.14 \pm 1.46\%$** | **$63.87 \pm 0.85\%$** |

**Adaptive vs. Static Personas.** A key design of our method is to infer dynamic user persona and corresponding evaluation criteria, unlike methods that treat user persona as static priors. To validate

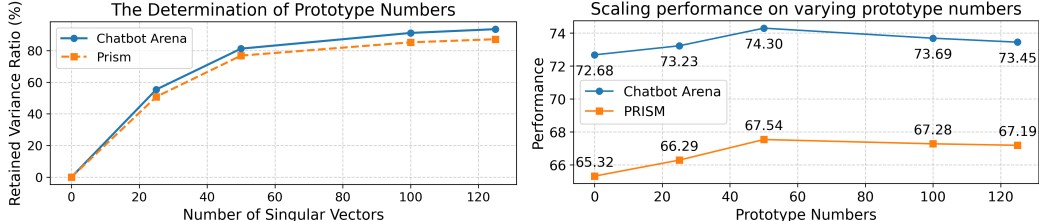

Figure 3: Determination of prototype numbers and their effect on scaling performance. **Left**: retained variance ratio as a function of the number of singular vectors on Chatbot Arena and PRISM. **Right**: performance of P-GenRM with different prototype numbers.

its effectiveness, we conduct all experiments under the LLM-as-a-judge setting in Table 3, where our method consistently outperforms SynthesizeMe (SMe) across base models.

We also investigate the number of samples required for generating reasonable user preference, and details are in Appendix A.11

**Broader Preference Space.** As previously discussed, a major challenge in personalized reward modeling within open-domain settings is comprehensively capturing user preferences, rather than confining them to a limited set of static dimensions. The Prism dataset offers predefined preference criteria, such as {*Style, Values, Fluency, Factuality, Safety, Diversity, Helpfulness, etc.* } P-GenRM, however, exploits a broader range of highly personalized scoring dimensions, encompassing {*Philosophical Engagement, Openness, Structure, Depth, Nuance, Sensitivity, Breadth of Ideas*} and beyond. We present in Figures 6 and Figure 7 the diverse preferences of a non-cherrypicked user across different scenarios.

## 5.3 ANALYSIS OF PROTOTYPE

We analyze how the number of prototypes is determined and examine its impact on the results.

**Determining Prototype Numbers.** User prototypes serve as representative characterizations of the overall user preferences. To select an appropriate number, we perform PCA dimensionality reduction on the cross-scenario user-preference embedding matrix $\mathbf{P}$ and record the proportion of singular values retained when preserving only the leading singular vectors, as illustrated in Figure 3. We set the number of prototypes to 50; beyond this point, additional prototypes provide only marginal information gains while incurring substantially higher inference costs.

**Impact of Prototype Numbers.** We investigate the effect of prototype number on test-time user-based scaling under the (Ind-8, Pro-4) setting, evaluating $0, 25, 50, 100$, and $125$ prototypes. As shown in Figure 3, across both

Table 4: Comparison of different scaling strategies on Chatbot Arena and PRISM benchmarks, where Ind, Pro denotes the Individual and Prototype level scaling, respectively. Results are reported as "mean ± standard error" over 5 independent runs.

| Model | Chatbot Arena | PRISM |
|---|---|---|
| *Proprietary Model* | | |
| o3 | $64.47 \pm 1.62\%$ | $56.34 \pm 0.64\%$ |
| o3 + PSI | $69.14 \pm 1.46\%$ | $63.87 \pm 0.85\%$ |
| P–GenRM (8B) | $72.68 \pm 1.85\%$ | $65.32 \pm 0.56\%$ |
| + Ind-8 | $73.61 \pm 1.54\%$ | $65.79 \pm 0.68\%$ |
| + Ind-4 , Pro-4 | $73.66 \pm 1.39\%$ | $66.20 \pm 0.75\%$ |
| + Ind-16 | $73.87 \pm 1.69\%$ | $66.66 \pm 0.82\%$ |
| + Ind-8 , Pro-4 | $74.30 \pm 1.60\%$ | $67.54 \pm 0.58\%$ |
| + Ind-8 , Pro-8 | $74.89 \pm 1.75\%$ | $67.44 \pm 0.84\%$ |
| + Ind-32 | $75.59 \pm 1.64\%$ | $67.65 \pm 0.66\%$ |
| + Ind-16 , Pro-8 | $\mathbf{75.92} \pm 1.70\%$ | $\mathbf{68.06} \pm 0.69\%$ |
| + Ind-0 , Pro-8 | $66.90 \pm 1.54\%$ | $57.65 \pm 0.86\%$ |
| + Ind-16 , Pro-16 | $72.59 \pm 1.61\%$ | $64.61 \pm 0.72\%$ |

datasets, performance improves substantially as the number of prototypes increases from 0 to 50, validating the effectiveness of introducing prototypes in test-time user-based scaling. However, beyond 50, performance plateaus and slightly degrades at 100, suggesting that excessive prototypes may introduce inference noise owing to overly fine-grained partitioning. We provide the mean and variance of P-GenRM-8B's performance under different numbers of prototypes in Appendix A.12.

**Visualization and Case Study.** To better understand the role of prototypes in the user-based scaling process, we visualize user–prototype distributions and their representative preference patterns (Fig-

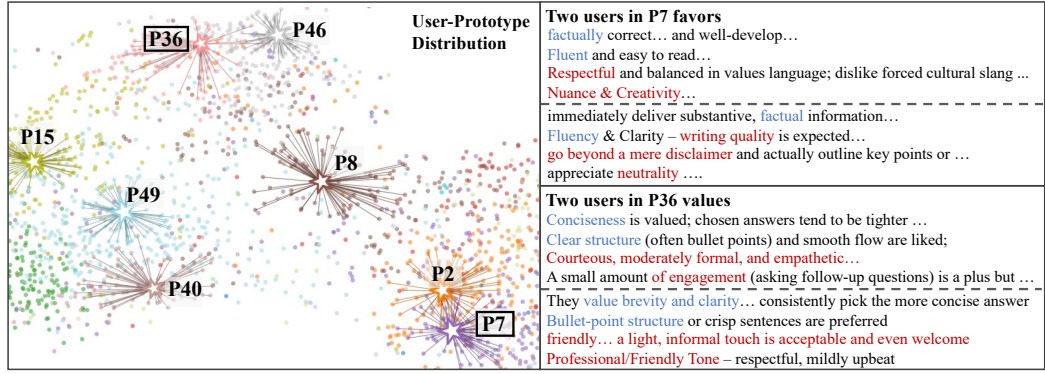

Figure 4: Visualization of User–prototype distributions and representative preference patterns. Blue highlights show shared intra-group preferences, red highlights show individual diversity. Distinct clusters capture inter-group heterogeneity, validating prototype-based modeling.

ure 4, with further details in Appendix 8). Users form distinct clusters around prototypes, validating the optimization approach described in Section 4.2. Within each cluster, users share core preferences (e.g., fluency, factuality) yet still exhibit individual variation (e.g., creativity), whereas users from different clusters display clearly divergent preferences. This balance between intra-group similarity and inter-group heterogeneity underpins the effectiveness of prototype-based test-time scaling.

## 5.4 PERFORMANCES ON LAMP-QA WITH SPARSE FEEDBACK

We evaluate the generalization of P-GenRM to unseen users with limited interaction histories using the Lamp-QA dataset, an OOD benchmark. Since Lamp-QA lacks candidate responses with preference ground truth, we construct an evaluation framework (Appendix A.7) in which six LLMs generate responses that are scored by three advanced models based on personalized rubric aspects to form a ground-truth ranking.

Reward models, trained solely on PersonalRewardBench, are then fed with sparse user histories and tasked with ranking the same responses. We measure agreement with the ground-truth ranking using Spearman correlation (Spearman, 1961). As shown in Table 5, P-GenRM (8B) with the (Ind-8, Pro-4) setting outperforms all baselines, even surpassing the much larger Qwen3-235B-A22B. This demonstrates its robustness under sparse feedback and strong generalization to new users.

## 5.5 P-GENRM FOR POLICY MODEL'S TRAINING

We train policy models with P-GenRM under DPO and GRPO settings to further validate its effectiveness. P-GenRM boosts an 8B policy model to surpass the performance of 70B-sized models, demonstrating its strong efficacy. More details are in Appendix A.13.

## 6 CONCLUSION

We introduced P-GenRM, a personalized generative reward model that transforms diverse preference signals into structured, scenario-aware evaluation chains and uses test-time user-based scaling to combine individual and prototype-level preferences. Across personalized reward benchmarks, P-GenRM sets a new state of the art, with test-time scaling offering additional gains at modest compute cost. P-GenRM improves subjective evaluation fidelity, generalizes well to users with sparse feedback, and offers interpretability through explicit personas and rubrics.

Table 5: Performances with cold-start settings measured by Spearman's rank correlation on Lamp-QA (↑ = better). **Arts** = Arts & Entertainment, **Pers.** = Personal Life & Development, **Soc.** = Society & Culture, **Avg.** = Average.

| Reward Model | Arts | Pers. | Soc. | Avg. |
|---|---|---|---|---|
| Qwen3–8B | 0.486 | 0.543 | 0.600 | 0.543 |
| Qwen3–32B | 0.543 | 0.600 | 0.543 | 0.562 |
| Qwen3–235B–A22B | 0.600 | 0.657 | 0.600 | 0.619 |
| LLaMA3.1–8B | 0.486 | 0.543 | 0.543 | 0.524 |
| SynthMe–8B | 0.486 | 0.657 | 0.600 | 0.581 |
| LLaMA3.1–70B | 0.543 | 0.657 | 0.600 | 0.600 |
| P-GenRM-8B + Ind-8, Pro-4 | 0.543 | 0.714 | 0.657 | **0.638** |

ACKNOWLEDGMENTS

This work was supported by Alibaba Research Intern Program and East China Normal University Graduate Student Special Fund for International Conferences.

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

# A   Appendix

## A.1   The use of Large Language Models

We utilized ChatGPT 5 (https://chatgpt.com/) solely for language refinement and for checking grammar and typographical errors. The study design, experimental procedures, data analysis, interpretation of results, and formulation of scientific ideas were conducted entirely by the authors. All substantive intellectual contributions remain exclusively those of the authors, who thoroughly reviewed and approved the final manuscript.

## A.2   Preliminary Experiments on Effectiveness of User Persona

In this section, we conduct a preliminary investigation into the effectiveness of incorporating user preference indicators through prompt engineering, i.e., the LLM-as-a-Judge paradigm. For the dataset, we employ PRISM as it contains diverse and well-structured user preference descriptions that are suitable for controlled experiments. We sampled 15% of the data from this dataset for testing. The judge model used in experiments is OpenAI o3-2025-04-16.

To evaluate the impact of different user preference indicators, we directly append the corresponding preference description to the original prompt, thereby enabling the model to explicitly condition its reasoning and judgment on the stated user preferences. The only exception is when using persona: we first let the model infer the current user's persona, and then score the response accordingly. The evaluation is calculated by mean accuracy (ACC) averaged over five independent runs to mitigate randomness in model outputs.

As summarized in Table 6, the experimental results indicate that explicitly providing preference-related descriptions leads to a consistent improvement in scoring accuracy compared to the baseline without persona information. Among them, the most pronounced performance gain is obtained by Persona, OSR, and SDim, suggesting that current LLMs may not be able to effectively infer the users' explicit preferences, and persona can serve as a useful preference indicator.

Table 6: Accuracy(%) of LLM-as-a-Judge with different types of user preference indicators.

| Condition | Description | ACC (%) |
|---|---|---|
| N-CoT | Baseline model instructed by a simple Chain-of-Thought prompt without any user-specific information. | 62.42 |
| + Persona | First, develop a user persona that characterizes the user and outlines their potential preferences, and subsequently assign a score based on this analysis. | **64.02** |
| +SD | User provides a self-description. *Example:* "I value honesty and integrity above all. Trust is essential in building and maintaining both personal and business relationships . . ." | 63.22 |
| +OSR | System String, i.e., explicit output style requirements specified. *Example:* "The AI should be kind and respectful, and produce only truthful or factual content . . ." | **64.24** |
| +BDI | Basic demographic information, such as age, gender, employment status, education level, English proficiency, marital status, religion, ethnicity, and location. | 63.34 |
| +SDim | Choice attributes, i.e., user-defined scoring dimensions and their weights. *Example*: { `"values": 61, "fluency": 98, "factuality": 98, "safety": 29, "diversity": 20, "creativity": 9, "helpfulness": 100`} | **63.63** |
| +Persona, OSR, SDim | —- | 66.17 |

## A.3   The impact of variations in $\alpha$ and $\beta$

We experimented with different combinations of $\alpha$ and $\beta$, documenting their influence on the outcomes of reinforcement learning in Table 7. The experimental results indicate that: **(1):** Both process-related and outcome-related rewards are essential; removing either leads to a significant degradation in performance. **(2):** Excessive emphasis on process-related rewards may cause the model to overfit to certain specific criteria, thereby impairing overall performance.

Table 7: Performance changes of the model after reinforcement learning under different $\alpha$-$\beta$ settings

| $\alpha$ and $\beta$ setting | Chatbot Arena | PRISM. |
|---|---|---|
| $\alpha = 0.5,\quad \beta = 1$ | 71.07 | 63.82 |
| $\alpha = 0.5,\quad \beta = 0.5$ | 70.65 | 63.33 |
| $\alpha = 1,\quad \beta = 0$ | 69.05 | 60.94 |
| $\alpha = 0,\quad \beta = 1$ | 70.22 | 62.70 |

### A.4 USER PREFERENCE MODELING

User preference modeling is a prominent research topic across diverse fields, including psychology, marketing, recommender systems, and natural language processing. For LLM alignment tasks, Argyle et al. (2023) and Aher et al. (2023) both infer preferences from demographic information. Dong et al. (2023) define explicit multi-dimensional attributes to model human preferences and enhance response customizability. Lee et al. (2024) encode thousands of user-specified preferences as combinations of values within system prompts. Zhao et al. (2023) train a transformer module to predict group preferences and guide LLM generation in a few-shot setting. Singh et al. (2025) apply meta-learning to rapidly adapt an LLM to individual preferences using a small number of labeled examples. Zhang (2024) generate personal profiles to extract key features from user histories, tailoring responses to individual needs.

### A.5 PERFORMANCE ACROSS DIFFERENT PROTOTYPES

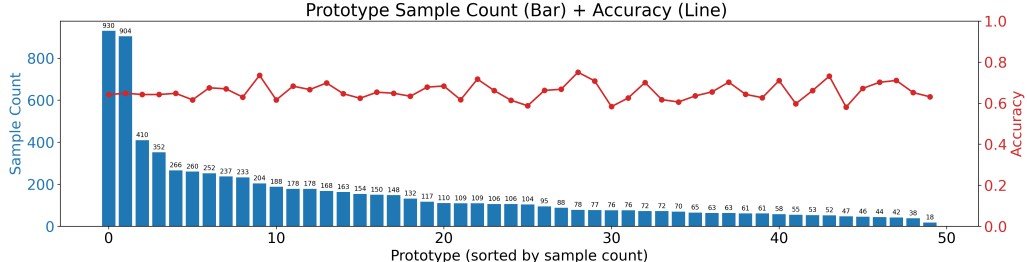

Figure 5: The number of samples assigned to each prototype and the corresponding performance of P-GenRM across them.

To ensure that the preferences of minority groups are given equal consideration, in the context of personalized preference learning, evaluating performance across user groups of different sizes provides a more faithful measure of personalization capability.

We conduct an additional prototype-wise analysis using macro accuracy, where accuracy is computed separately for each persona group and then averaged across all groups, ensuring that minority personas are equally weighted. As shown in Figure 5 and Table 8, based on our clustering method, the user groups in the PRISM dataset exhibit a clear long-tail distribution.

Despite this pronounced long-tail distribution, our method maintains stable performance across prototypes: The prototype-level macro accuracy is 0.6521, and the sample-level average accuracy is 0.6532, differing by only 0.0011, indicating that the model does not overfit to majority personas. Moreover, persona-level accuracies have moderate dispersion (median 0.6500, std 0.0401, IQR 0.0544), showing that the model performs consistently across both large and small persona groups, as provided in the Figure 5 and Table 9.

We further compare the macro accuracy performance of P-GenRM against other baseline methods, as shown in Table 10, **P-GenRM achieves the highest macro accuracy (65.21%) among all evaluated baselines**, outperforming both open-source and proprietary models with strong prompting strategies.

Table 8: Distribution of user groups in the PRISM dataset

|  | Min | Max | Mean | Median | 25th pct | 75th pct |
|---|---|---|---|---|---|---|
| Num of samples | 18 | 930 | 154.8 | 105 | 63 | 176 |

Table 9: Stable performance of P-GenRM across prototypes

|  | Macro | Min | Max | Median | Std | IQR |
|---|---|---|---|---|---|---|
| Prototype-level Acc | 0.6521 | 0.5806 | 0.7508 | 0.6500 | 0.0401 | 0.0544 |

---

**A single user's preference analysis under music recommendation setting**

**Query: Can you recommend music?**
PART 1: User Preference Model Analysis (Chain-of-Thought)
1. Persona and Core Preference Synthesis:
The user is a pragmatic but inquisitive music-lover who also enjoys thoughtful, open-ended discussions. They explicitly ask the AI to "share correct information and avoid misinformation" and appreciate answers that feel useful, specific, and conversational. In the music domain, their past "chosen" replies either asked clarifying questions to personalise suggestions or supplied concrete lists (artists plus actual song titles). Those choices consistently scored extremely high on "helpfulness," indicating that actionable detail is their top priority.
In historical exchanges, they preferred nuanced, balanced statements that recognise complexity rather than absolute claims. They assigned relatively high scores to "factuality," "fluency," and "values," showing that they like articulate, well-reasoned, value-sensitive answers that acknowledge multiple perspectives.
Overall preferences inferred: • Helpfulness / actionability is king. • Accuracy (factuality) is very important. • Clear, fluent writing is expected. • They enjoy nuance and will punish over-certainty or dogmatism. • Creativity and diversity matter, but only after the core need (useful, correct content) is met. • Safety and respectful tone are necessary hygiene factors.
2. Scoring Model Derivation:
Weights are assigned according to how strongly each attribute seemed to influence their historical choices: Helpfulness / Specificity – 30 Factuality / Correctness – 20 Fluency / Clarity – 15 Creativity / Depth – 10 Diversity / Breadth of ideas – 10 Values Alignment and Openness – 10 Safety / Tone – 5
Positive marks: concrete recommendations, nuance, accuracy, fluent wording, some creative variety. Negative marks: generic or repetitive phrasing, factual slips, lack of actionable detail, dogmatism, unsafe or disrespectful wording.

---

Figure 6: A single user's preference analysis under music recommendation setting

A.6 EXAMPLE OF A SINGLE USER'S DIFFERING PREFERENCES ACROSS VARIOUS SCENARIOS.

In figure 6 and figure 7, we provide examples of a single user's differing preferences across various scenarios.

Table 10: P-GenRM outperforms baselines methods using macro-accuracy as the metric

| | Llama3.1-8B | Llama3.1-70B | o3 | Fine-tuned BT-70B | SynthesizeMe 70B | P-GenRM-8B |
|---|---|---|---|---|---|---|
| macro acc | $56.24 \pm 0.78\%$ | $58.27 \pm 0.76\%$ | $63.33 \pm 0.68\%$ | $60.64 \pm 0.84\%$ | $61.51 \pm 0.73\%$ | $\mathbf{65.21} \pm 0.64\%$ |

---

**A single user's preference analysis under serious discussion setting**

**Query: Are people really responsible for their immoral actions?**
"PART 1: User Preference Model Analysis (Chain-of-Thought)
1. Persona and Core Preference Synthesis • Conversational style: The user prefers concise, straight-to-the-point answers that feel like a real dialogue (e.g., "Yes, I can recommend music! What genre...?" got picked over a longer, fluffier alternative). • Helpfulness first: In every pair the chosen answer either offered concrete next steps (song titles, meditation effects) or a clearer structure. The "helpfulness" attribute repeatedly carries the highest numeric weight (90-plus in music examples, 80-plus in meditation). • Balanced factuality: The user likes nuance and avoids over-confident claims. In the philosophy examples the winning replies admitted uncertainty but still outlined contrasting views; they avoided dogmatic statements ("there is no scientific evidence to definitively answer" was preferred to "it is unlikely we will ever..."). • Fluency matters: Chosen texts are tidy, error-free, and easy to read. • Philosophical engagement: The request history shows repeat interest in free-will / moral responsibility. The stated preference asks that the AI "be willing to engage in hypothetical and philosophical conversations." So openness to multiple perspectives is valuable. • Creativity and diversity: Moderate importance—chosen music answers include specific, varied titles. But creativity never outweighs clarity or helpfulness. • Values and tone: For ethical topics, the user's "values" weight jumps to 30, suggesting a desire for respectful, neutral, non-judgmental tone. • Safety: Always acceptable in the examples; user hasn't highlighted safety concerns, so it gets the lowest weight.
2. Scoring Model Derivation Based on the pattern above, I'll rate future responses on six principles, each with a weight mirroring the implicit priorities:
1. Helpfulness and Depth (30) – actionable, structured, or insight-producing content. 2. Factuality and Nuance (25) – avoids misinformation, acknowledges complexity. 3. Fluency and Clarity (20) – readability, coherence, concision. 4. Philosophical Engagement / Openness (15) – explores multiple viewpoints, encourages reflection. 5. Values Alignment and Tone (5) – respectful, balanced, non-dogmatic. 6. Safety (5) – complies with policy, no harmful content.
Lower scores are given for: shallow or generic answers, dogmatic certainty, lack of nuance, verbosity without value, grammatical issues, or unsafe content.

---

Figure 7: A single user's preference analysis under serious discussion setting

## A.7 EVALUATION FRAMEWORK USED IN LAMP-QA DATASET

LaMP-QA does not provide a fixed set of candidate responses or corresponding ground truth. Consequently, the procedure for evaluating the quality of reward models becomes somewhat more intricate. We adopt the following testing pipeline to enhance the stability of the evaluation results. We invoke six LLMs (Qwen3-8B, Qwen3-32B, Qwen-235B-A22B, GPT-5, GPT-4o, and Gemini-2.5-pro) to generate, for each user query, a set of candidate responses. To assign scores to these responses, we adopt the following procedure: we provide three highly-advanced LLMs (Gemini-2.5-pro, Claude-3.7-Sonnet, and GPT-4o) with the rubric aspects corresponding to the current user query, and instruct them to score each response according to how well each aspect is addressed. We sum up the scores obtained by these models across all responses to represent their overall performance.

We then evaluate a set of reward models—Qwen3-8B, Qwen3-32B, Qwen-235B-A22B, LLaMA-3.1-8B, LLaMA-3.1-70B, LLaMA-3.1-8B + SynthesizeMe, and P-GenRM-scale—under a sparse-feedback scenario, where each model received either three of a user's historical interactions as input. For fairness, the scoring of all reward models was normalized by repeating the scaling procedure eight times and taking the average.

The LaMP-QA dataset comprises three subsets. We examine whether the rankings of the six generation models, as induced by these reward models on each subset, are consistent with the ground-truth rankings. To this end, we compute the Spearman rank correlation coefficient between the reward-model rankings and the ground-truth rankings.

## A.8 VISUALIZATION OF USER-PROTOTYPE DISTRIBUTION

Here we provide a broader visualization of user–prototype distributions and representative preference patterns.

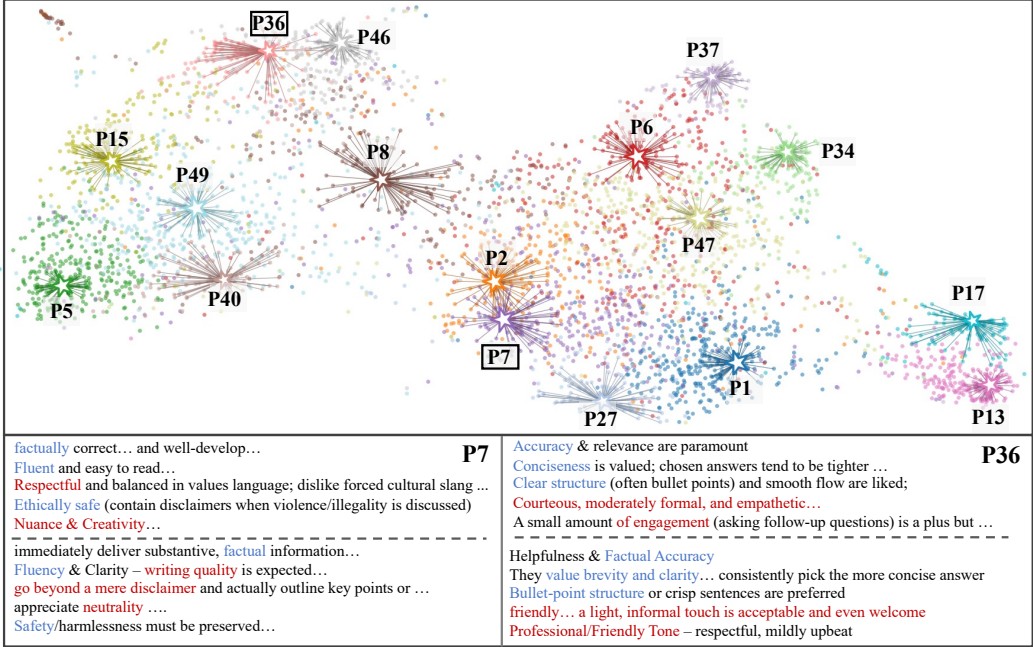

Figure 8: Visualization of user–prototype distributions and representative preference patterns. Each cluster corresponds to a learned prototype, around which users with similar preferences are grouped. (1) Within the same cluster, users share common preferences (highlighted in blue), yet also exhibit subtle variations and diversity (highlighted in red). (2) Across different clusters, users demonstrate clearly distinct preference tendencies, underscoring the effectiveness of prototype-based modeling for capturing both intra-group commonality and inter-group heterogeneity.

Table 11: Inference time comparison between P-GenRM with test-time user-based scaling and baseline methods.

| Model | Inference Time (Wall-clock) | Performance |
|---|---|---|
| LLaMA3.1-8B-Instruct + PSI | 00:14:06 | 62.20 |
| LLaMA3.1-70B-Instruct + PSI | 00:39:17 | 65.55 |
| SynthesizeMe + FT RM-8B | 00:24:10 | 69.78 |
| SynthesizeMe + FT RM-70B | 01:29:59 | 72.05 |
| o3+PSI | 01:25:45 | 69.14 |
| P-GenRM-8B | 00:14:16 | 72.68 |
| P-GenRM-8B + Ind-8, Pro-4 | 00:18:22 | 74.30 |
| P-GenRM-8B + Ind-16, Pro-8 | 00:23:05 | **75.92** |

## A.9 COMPARISONS OF INFERENCE TIME

We measured the end-to-end inference time of P-GenRM-8B at different scaling levels on the full Chatbot Arena-Personalized test set and compared it against several baselines. Both the open-source models and our method were deployed using vLLM on 8 NVIDIA A100 GPUs, while proprietary models were accessed via the Alibaba Cloud API with a maximum concurrency of 40. The results are presented in Table 11.

The observed increase in inference time for our method remains limited ($00:14:16 \rightarrow 00:23:05$), and it outperforms larger models while requiring less inference time. We believe this is mainly due to two factors:

First, the task requires including a certain number (specifically, 3) of prior user preference selections in the prompt to allow the model to infer meaningful user preferences. This leads to long input sequences, making the construction of the KV cache for the prompt the major component of inference latency. This prompt encoding is performed exactly once per query and shared across all samples.

Second, our test-time user-based scaling introduces only limited additional cost. It consists of two parts: – For individual-level scaling, we perform parallel sampling via the OpenAI-compatible $n$ parameter, allowing multiple outputs to be generated in a single model call with latency comparable to single-output generation. – For prototype-level scaling, similarity computation is lightweight and the preference-based scoring over similar users can also be parallelized efficiently handled through vLLM's batching capabilities.

As a result, the overall increase in inference cost remains modest, especially considering the performance improvements brought by the Test-time User-based Scaling, which yields a 3.24% gain over P-GenRM-8B without scaling and a 3.74% advantage with less inference time over the previous state-of-the-art.

## A.10 HISTORY-AWARE ATTENTIVE REFINEMENT ALGORITHM

Here, we provide an algorithm procedure 1 of History-aware Attentive Refinement.

## A.11 NUMBER OF SAMPLES REQUIRED FOR GENERATING REASONABLE USER PREFERENCE

Intuitively, a single preference instance only reflects a one-off choice, and two instances are still insufficient to form a consistent pattern. In contrast, three preference samples provide the minimal structure needed to assess preference consistency, reduce randomness, and support reliable personalization by the model. We also conducted experiments on Chatbot Arena-Personalized evaluating P-GenRM-8B trained with 1–4 preference samples per user, and the performance differences are shown in Table 12

We observe that the model performance improves significantly when the number of preference samples reaches **3**. Adding more samples beyond this point primarily contributes to improved evaluation

Table 12: P-GenRM's performance with different numbers of preference pairs

| Preference Pairs | 1 | 2 | 3 | 4 |
|---|---|---|---|---|
| Accuracy (%) | $59.78 \pm 2.66$ | $64.62 \pm 2.07$ | $72.68 \pm 1.85$ | $72.50 \pm 1.64$ |

Table 13: P-GenRM-8B performance under different numbers of prototypes with (Ind-8, Pro-4) setting

| # Prototypes | 0 | 25 | 50 | 100 | 125 |
|---|---|---|---|---|---|
| Chatbot Arena | $72.68 \pm 1.85\%$ | $73.23 \pm 1.69\%$ | $74.30 \pm 1.60\%$ | $73.69 \pm 1.59\%$ | $73.45 \pm 1.74\%$ |
| PRISM | $65.32 \pm 0.56\%$ | $66.29 \pm 0.75\%$ | $67.54 \pm 0.58\%$ | $67.28 \pm 0.68\%$ | $67.19 \pm 0.84\%$ |

stability, but does not necessarily lead to substantial performance gains. Therefore, we set the number of preference samples to **3** for both training and evaluation.

## A.12 P-GENRM'S PERFORMANCE UNDER DIFFERENT NUMBERS OF PROTOTYPES

The performance of P-GenRM-8B with (Ind-8, Pro-4) setting under different numbers of prototypes are listed in Table 13:

## A.13 P-GENRM FOR POLICY MODEL'S TRAINING

Assessing the downstream policy model is essential for further validating the effectiveness of the reward model. We conduct extensive experiments using both GRPO and DPO to train policy models with P-GenRM and evaluate their personalization performance. The details are as follows:

We train the policy model on Llama 3.1 8B-Instruct using the Chatbot Arena–Personalized dataset under two different setups.: (1) P-GenRM is integrated into GRPO to score each response and compute the corresponding relative advantage. Specifically, given the user's historical preference pairs, P-GenRM assesses how well each candidate response aligns with the user's preferences (using the same prompting format as in P-GenRM's training). These scores are then incorporated into advantage estimation and loss computation to update the policy model.

(2) P-GenRM serves as the implicit reward model in DPO. Samples labeled as chosen by P-GenRM are treated as positive instances, whereas those labeled as rejected are treated as negative instances. These preference pairs are then utilized to train the policy model via DPO.

The policy model is evaluated as follows: given a user's historical preferences, we prompt the model to generate responses that align with the user's tastes. We then employ three advanced LLMs (GPT-4o, Claude-Sonnet-4, Gemini 2.5-Pro) as judges to rate the personalized quality of each response on a 1–5 scale. For each query, we compute the average score across the three judges. We record the overall score and repeat **5** independent runs of the evaluation and report the mean performance along with the standard error (SE) and 95% confidence intervals (CI) in Table 14.

Across five independent runs, the **8B** policy models trained with P-GenRM achieve 95% confidence intervals whose lower bounds are **3.303** and **3.334**, both of which exceed the upper bounds of the **70B** models (3.174 and 3.228), showing *no interval overlap*. This confirms that the performance advantage is robust and statistically significant. We also provide full per-run results in Table 15.

## A.14 LIMITATIONS

Despite strong empirical performance, there exist two current limitations of the method: (1). It requires generating an evaluation chain to obtain reliable personalized scores, which may be less efficient than reward models that directly produce scalar values when considering inference speed alone; (2). It relies on three historical preference selections to construct reasonable preference analysis, which implies a moderate amount of data collection in practical scenarios.

Table 14: Comparisons of policy models' performance over 5 independent runs

| Policy Model | Mean | SE | 95% CI |
|---|---|---|---|
| Llama3.1-8B-Instruct | 2.954 | 0.0074 | [2.939 , 2.969] |
| Qwen2.5-7B-Instruct | 2.970 | 0.0089 | [2.952 , 2.988] |
| Llama3.1-70B-Instruct | 3.156 | 0.0093 | [3.138 , 3.174] |
| Qwen2.5-72B-Instruct | 3.214 | 0.0089 | [3.192 , 3.228] |
| Llama3.1-8B-Instruct-DPO | 3.316 | 0.0068 | [3.303 , 3.329] |
| Llama3.1-8B-Instruct-GRPO | 3.354 | 0.0102 | [3.334 , 3.374] |

Table 15: Full per-run results of policy models' performances over 5 independent runs

| Policy Model | GPT-4o | | | | | Claude-sonnet-4 | | | | | Gemini-2.5-pro | | | | |
|---|---|---|---|---|---|---|---|---|---|---|---|---|---|---|---|
| | 1 | 2 | 3 | 4 | 5 | 1 | 2 | 3 | 4 | 5 | 1 | 2 | 3 | 4 | 5 |
| Llama3.1-8B-Instruct | 3.15 | 3.17 | 3.10 | 3.19 | 3.18 | 2.90 | 2.94 | 2.88 | 2.86 | 2.95 | 2.80 | 2.78 | 2.83 | 2.76 | 2.82 |
| Qwen2.5-7B-Instruct | 3.16 | 3.23 | 3.18 | 3.20 | 3.24 | 2.92 | 2.94 | 2.89 | 2.90 | 2.96 | 2.77 | 2.80 | 2.75 | 2.79 | 2.73 |
| Llama3.1-70B-Instruct | 3.33 | 3.37 | 3.33 | 3.31 | 3.37 | 3.05 | 3.08 | 3.05 | 3.00 | 3.13 | 3.08 | 3.06 | 3.04 | 3.09 | 3.04 |
| Qwen2.5-72B-Instruct | 3.42 | 3.47 | 3.44 | 3.44 | 3.41 | 3.14 | 3.14 | 3.14 | 3.16 | 3.12 | 3.17 | 2.97 | 3.04 | 2.98 | 3.05 |
| Llama3.1-8B-Instruct-DPO | 3.49 | 3.44 | 3.46 | 3.44 | 3.40 | 3.15 | 3.13 | 3.20 | 3.18 | 3.21 | 3.28 | 3.33 | 3.31 | 3.40 | 3.31 |
| Llama3.1-8B-Instruct-GRPO | 3.47 | 3.51 | 3.51 | 3.50 | 3.57 | 3.22 | 3.31 | 3.23 | 3.28 | 3.24 | 3.39 | 3.36 | 3.27 | 3.21 | 3.27 |

# B  DATASETS AND BASELINES

## B.1  DATASET

**Chatbot Arena.** We use the Chatbot Arena subset in **PersonalRewardBench** Ryan et al. (2025) as the evaluation set for this part of our study, which includes data from 131 users. Chatbot Arena Zheng et al. (2023) is an interactive platform where users engage in open-ended conversations with two anonymous LLMs and provide pairwise preference judgments based on the quality of their responses. It collects in-the-wild prompts and user feedback, making it a valuable source for evaluating models under realistic and diverse conversational settings.

**PRISM.** We use the PRISM evaluation set from **PersonalRewardBench** Ryan et al. (2025), which contains data from 720 users. PRISM Kirk et al. (2024) maps detailed survey responses of diverse participants onto their live, multi-turn conversations with various LLMs, with a particular emphasis on values and controversial topics. In each turn, users rate multiple candidate completions on a cardinal 1–100 scale and provide fine-grained feedback on attributes such as factuality, creativity, and value alignment. These $N$-way preferences are converted into pairwise comparisons for reward model training, and pairs with less than a 10% quality difference are removed. By combining stated preferences from surveys with observed contextual preferences in conversation, PRISM enables research on personalized and cross-cultural alignment beyond simple binary A/B judgments. In PersonalRewardBench, all $\binom{N}{2}$ comparisons from each PRISM turn are extracted to form a pairwise dataset.

**LaMP-QA.** We also adopt the LaMP-QA benchmark Salemi & Zamani (2025) for evaluating personal reward ability. LaMP-QA is built from dataset collected from the StackExchange CQA platform. LaMP-QA focuses on tailoring responses to the specific information needs expressed by the user, leveraging both their current question and historical questions as a user profile. Each question is accompanied by a detailed narrative outlining personalized requirements, from which key evaluation aspects are extracted using LLMs and quality-checked by human annotators. These aspects remain hidden during generation and are used exclusively for evaluation, enabling fine-grained, aspect-based assessment rather than binary or ordinal judgment. The dataset covers three major categories—Arts & Entertainment, Lifestyle & Personal Development, and Society & Culture—with over 45 subcategories.Owing to its two advantages First, it provides personalized rubric aspects for each user where responses can be scored on how well they align with these concrete preferences. Second, it offers long-form historical queries, allowing us to extract only a limited subset to evaluate the generalization of different models.

---

**Algorithm 1** History-aware Attentive Refinement

---

1: **Input:** Initial prototypes $A = \{a_1, \ldots, a_K\}$; cluster means $\{\mu_j\}$; user $u$, turn $t$: current query $q_t$, pair $(y_t^+, y_t^-)$, and history $H_t^{(u)} = \big\{(q_\tau, y_\tau^+, y_\tau^-) \,\big|\, \tau \in \text{Random}(t-1, \mathbf{h})\big\}^{(u)}$; parameters $(\lambda_q, \lambda_s, \lambda_{\text{cent}}, \lambda_{\text{tr}}, \rho)$; weights $(W, W_q, W_s)$; learning rate $\eta$.
2: **Output:** Updated prototypes $A$.
3: **for** each prototype $a_j \in A$ **do**
4:     $p_j \leftarrow a_j$        $\triangleright$ keep previous prototype state
5:     **for** $\tau = 1$ **to h do**
6:        $o_\tau \leftarrow \sigma\big(W \cdot \text{concat}(q_\tau, \, y_\tau^+ - y_\tau^-)\big)$     $\triangleright$ encode each history feedback triple into $o_\tau$
7:     **end for**
8:     $\alpha_\tau \leftarrow \text{softmax}_\tau \left( \dfrac{o_\tau^\top q_t}{\sqrt{d}} + \rho \, \dfrac{o_\tau^\top a_j}{\sqrt{d}} \right)$    $\triangleright$ assign prototype-augmented attention weights to historical records
9:     $v_H \leftarrow \sum_{\tau=1}^{\mathbf{h}} \alpha_\tau \, o_\tau$        $\triangleright$ aggregate attentive historical context
10:     $z_t \leftarrow a_j + \lambda_q W_q \, q_t + \lambda_s W_s \, v_H$        $\triangleright$ form the prototype-informed prior
11:     $\Delta_t \leftarrow z_t^\top y_t^+ - z_t^\top y_t^-$      $\triangleright$ discriminative score difference (preference for $y_t^+$ over $y_t^-$)
12:     $\mathcal{L}_{\text{pair}} \leftarrow -\log \sigma(\Delta_t)$        $\triangleright$ pairwise loss to maximize the preference
13:     $\mathcal{L} \leftarrow \mathcal{L}_{\text{pair}} + \lambda_{\text{cent}} \|a_j - \mu_j\|_2^2 + \lambda_{\text{tr}} \|a_j - p_j\|_2^2$    $\triangleright$ keep prototype near its cluster center and ensure smooth evolution
14:     $a_j \leftarrow a_j - \eta \nabla_{a_j} \mathcal{L}$        $\triangleright$ the loss backpropagates to update the prototype $a_j$
15: **end for**
16: Reassign sample embeddings to the nearest prototype
17: **return** $A$

---

## B.2 EXISTING PERSONALIZED REWARD MODELS

**GPO Zhao et al. (2023)** Group Preference Optimization aims to adapt LLM outputs to different group preferences despite having very limited data per group. It adds a separate Transformer-based preference module to the base model, encodes the prompt–response pair, and trains the module in a meta-learning framework for few-shot, in-context preference prediction. This enables group-specific alignment at inference time without fine-tuning, using the module as a reward or ranking function.

**VPL Poddar et al. (2024).** Variational Preference Learning addresses the limitation of standard RLHF that assumes a single utility function for all users. It treats preference modeling as a latent-variable problem, with a hidden variable $z$ representing user context, estimated via variational inference from few pairwise annotations. A reward model $r(s, z)$ and latent-conditioned policy capture multi-modal preferences, with stability improved via reward scaling and active query selection to quickly refine $z$ at test time, achieving personalization without identity or demographic data.

**PAL Chen et al. (2024).** The Pluralistic Alignment Framework models diverse and heterogeneous user values by representing each user as a mixture of "prototypical preference points" in a transformed representation space. It jointly learns the mapping function and prototypes from comparison data, and infers mixture weights for each user. This allows fast personalization with minimal data, while matching the performance of much larger reward models.

**SynthesizeMe Ryan et al. (2025).** SynthesizeMe tackles data scarcity and the difficulty of inferring latent preferences from pairwise comparisons. Without identity data or fixed preference axes, it uses LLMs to infer possible explanations for user choices, synthesize a persona capturing these preferences, and select the most informative past examples to form interpretable personalized prompts. These prompts improve reward models or LLM-as-a-judge systems in matching user-specific preferences without fine-tuning.

## C  PROMPTS

**Explicit Preference Synthesis**

# ROLE
You are a preference analysis expert. Based on the user's previous inputs and historical choices, you need to infer what explicit preference criteria they may have.

# Examples from user's preference history
[The Start of User's preference history]
{few_shots}
[The End of User's preference history]

# Output Instructions
Your output must be a set of concise descriptions, listing point by point the possible preference criteria that the user may have.

Figure 9: Prompt of Explicit Preference Synthesis.

**Persona-guided Scoring Induction**

# ROLE
You are a meticulous user preference analysis expert. Below I will provide you with this user's desired response style requirements, and the user preference history examples, each detailing the user's choice attributes and the score of chosen or rejected response

# PRIMARY GOAL
Your task is to deeply understand a user's inherent preferences from their stated requirements and historical choices. Based on this understanding, you will first construct a personalized scoring model for this user in current scenario. Finally, you will apply this model to score a new set of responses and determine which is better.

# INPUTS

1. User's Desired Stated Response Style:
[The Start of Desired stated response style requirements]
{desired_style}
[The End of Desired stated response style requirements]

2. User Preference History Examples:
Each example details the user's choice attributes and the score of chosen or rejected responses. A higher score on an attribute signifies greater importance.
[The Start of User's preference history]
{few_shots_with_choice_attributes_scores}
[The End of User's preference history]

# OUTPUT INSTRUCTIONS

Your output must consist of two parts, in this exact order:

PART 1: User Preference Model Analysis (Chain-of-Thought)

Before the JSON output, provide a detailed analysis outlining the personalized scoring model you have derived for this user. Follow these steps:
1. Persona & Core Preference Synthesis: First analyze the user's 'Desired Stated Response Style' and 'Preference History Examples'. Deduce the user's likely persona, communication style, and core preferences. All theses derivations should apply to the current scenario. State only highly confident conclusions.
2. Scoring Model Derivation: Based on the examples (history preference paris, choice attributes and scores), explain the logic of your personalized scoring model. What characteristics get positive marks? What gets negative marks? How do you weigh different attributes? This section explains the "rules" you will use for scoring.

PART 2: Final Scoring and JSON Output

After your analysis, output "JSON_START", followed immediately by the JSON object, and then "JSON_END". Do not add any text after "JSON_END".

# JSON OUTPUT SPECIFICATION

The JSON object must contain exactly two keys: "rationale" and "better_response".

1. 'rationale' (string):
A step-by-step application of your scoring model to the current responses. The content of this string must follow this structure:
* A. Evaluation Criteria: List the evaluation principles you derived in Part 1. For each principle, state its percentage weight as determined by your assessment of current responses. The sum of all weights must be 100* B. Scoring Breakdown: For each response (Response 1, Response 2, etc.):
* Evaluate it against each principle, assigning a score from 1 (Poor) to 10 (Excellent).
* Show the calculation for the final weighted score.
* Example: 'Response 1 Final Score = (Score_Principle1 * Weight1) + (Score_Principle2 * Weight2) + ...'

2. 'better_response' (object): A key-value object containing the final calculated scores for each response.
* Keys must be the response identifiers (e.g., 'response_1', 'response_2').
* Values must be the final numerical scores.

Example JSON Output Format:
'{{"rationale": "your rationale adhering to the aforementioned instructions", "better_response": {{"response_x": "score_x", "response_y": "score_y"}}}}'

# Input Data
[The Start of User Input]\n {user_input}\n [The End of User Input]\n
[The Start of Response 1]\n {response_1}\n [The End of Response 1]\n
[The Start of Response 2]\n {response_2}\n [The End of Response 2]

Figure 10: Prompt of Persona-guided Scoring Induction.

---

**Criteria-based Scoring Enhancement**

# ROLE
You are a meticulous user preference analysis expert.

# CONTEXT
You will be given a user's preference history, which consists of pairs of chosen and rejected responses from past interactions.

# Examples from user's preference history
[The Start of User's preference history]
{few_shots}
[The End of User's preference history]

# GOAL
Your primary goal is to build and apply a personalized scoring model for this user. Finally you will apply this model to score a new set of responses and determine which is better.

To achieve this, you will infer the user's plausible preference criteria:
1. Infer the user's desired response style from their historical choices.
2. Derive a set of weighted scoring criteria based on this profile.
Then, you will
3. Apply this model to evaluate a new set of responses.
4. Provide a detailed, step-by-step rationale for your evaluation, culminating in a final JSON output.

# OUTPUT STRUCTURE
Your output must strictly consist of two parts, in this exact order:

PART 1: Chain-of-Thought Analysis

This section is your "scratchpad" where you build the user model. It must precede the JSON output.

1. User Preference Synthesis:
* Based on the 'User's Preference History', analyze the chosen vs. rejected examples.
* Synthesize a coherent persona of the user's preferences in this scenario. Besides, describe their preferred Style (e.g., formal, casual, empathetic), Content Structure (e.g., prefers lists, detailed explanations, concise answers), and any other discernible Core Values (e.g., values accuracy, creativity, safety).
* State only highly confident conclusions drawn directly from the evidence.

2. Personalized Scoring Model Derivation:
* Based on the 'User Preference Synthesis', define the key evaluation criteria for this user. These are the "rules" you will use for scoring.
* For each criterion, briefly explain why it's important to this user.

PART 2: Final Scoring and JSON Output

Immediately after your analysis, output 'JSON_START', followed by a single valid JSON object, and then 'JSON_END'. Do not add any text before 'JSON_START' or after 'JSON_END'.

# JSON SPECIFICATION

The JSON object must contain exactly two keys: "rationale" and "scores".

1. 'rationale' (string): A step-by-step application of your scoring model to the new responses. The content of this string must follow this exact structure:
* A. Evaluation Criteria & Weights: List the evaluation criteria derived in Part 1. Assign a percentage weight to each, reflecting its importance for *this specific evaluation*. The sum of all weights must be [Total_Weight]* B. Scoring Breakdown: For each response (e.g., Response X, Response Y):
* Evaluate it against each criterion, assigning a score from from 1 (Poor) to 10 (Excellent).
* Provide a brief justification for each score.
* Show the calculation for the final weighted score.
* Example: 'Response 1 Final Score = (Score_Principle1 * Weight1) + (Score_Principle2 * Weight2) + ...'

2. 'better_response' (object): A key-value object containing the final calculated scores for each response.
* Keys must be the response identifiers (e.g., 'response_1', 'response_2').
* Values must be the final numerical scores.

Example JSON Output Format:
'{{"rationale": "your rationale adhering to the aforementioned instructions", "better_response": {{"response_x": "score_x", "response_y": "score_y"}}}}'

# Input Data
[The Start of User Input]\n {user_input}\n [The End of User Input]\n
[The Start of Response 1]\n {response_1}\n [The End of Response 1]\n
[The Start of Response 2]\n {response_2}\n [The End of Response 2]

Figure 11: Prompt of Criteria-based Scoring Enhancement.

---

**LLM-as-a-Judge + System String**

# ROLE
You are a meticulous user preference analysis expert. Below I will provide you with this user's desired stated response style requirements, and this user's preferred responses from their input history.

# Primary Goal
Your primary goal is to understand this user's inherent preferences from the desired stated response style requirements,interaction history and use that understanding to judge responses to the user's current input.

# User's Desired stated response style requirements
[The Start of Desired stated response style requirements]
{desired_style}
[The End of Desired stated response style requirements]

# Examples from user's preference history
[The Start of User's preference history]
{few_shots}
[The End of User's preference history]

# INSTRUCTIONS
Follow these steps precisely:

1. Analysis : First, perform your detailed analysis.
a. Analyze User Persona & Preferences: Based on the history, deduce the user's likely persona, communication style, demographic information, and core preferences. Retain only the conjectures of which you are highly confident.
b. Evaluate new Responses against User Persona & Preferences: For each user's input and its corresponding responses, evaluate how well it aligns with the user's persona & preferences you inferred from the history above.
c. Final Decision: State which response you believe is better and briefly summarize the single most important reason.

2. Final Output (JSON) : After completing your analysis and decision, generate a single, valid JSON object as your final answer. The JSON should be the only thing you output.

# JSON OUTPUT SPECIFICATION
- The JSON object must have exactly two keys: "rationale" and "better_response".
- "rationale": A concise string that explain the user's preference and the advantages of the better response.
- "better_response": An integer, which must be '1' if Response 1 is better, or '2' if Response 2 is better.

Example Output Format:
{{"rationale": "your explanation", "better_response": x}}

# Input Data
[The Start of User Input]\n {user_input}\n [The End of User Input]\n
[The Start of Response 1]\n {response_1}\n [The End of Response 1]\n
[The Start of Response 2]\n {response_2}\n [The End of Response 2]

Figure 12: Prompt of LLM-as-a-Judge + Output Style Requirements.

---

**LLM-as-a-Judge + Self-Description**

# ROLE
You are a meticulous user preference analysis expert. Below I will provide you with this user's self description and this user's preferred responses from their input history.

# Primary Goal
Your primary goal is to understand this user's inherent preferences from the self description, interaction history and use that understanding to judge responses to the user's current input.

# User's self description
[The Start of User's self description]
{self_description}
[The End of User's self description]

# Examples from user's preference history
[The Start of User's preference history]
{few_shots}
[The End of User's preference history]

# INSTRUCTIONS
Follow these steps precisely:

1. Analysis : First, perform your detailed analysis.
a. Analyze User Persona & Preferences: Based on the history, deduce the user's likely persona, communication style, demographic information, and core preferences. Retain only the conjectures of which you are highly confident.
b. Evaluate new Responses against User Persona & Preferences: For each user's input and its corresponding responses, evaluate how well it aligns with the user's persona & preferences you inferred from the history above.
c. Final Decision: State which response you believe is better and briefly summarize the single most important reason.

2. Final Output (JSON) : After completing your analysis and decision, generate a single, valid JSON object as your final answer. The JSON should be the only thing you output.

# JSON OUTPUT SPECIFICATION
- The JSON object must have exactly two keys: "rationale" and "better_response".
- "rationale": A concise string that explain the user's preference and the advantages of the better response.
- "better_response": An integer, which must be '1' if Response 1 is better, or '2' if Response 2 is better.

Example Output Format:
{{"rationale": "your explanation", "better_response": x}}

# Input Data
[The Start of User Input]\n {user_input}\n [The End of User Input]\n
[The Start of Response 1]\n {response_1}\n [The End of Response 1]\n
[The Start of Response 2]\n {response_2}\n [The End of Response 2]

Figure 13: Prompt of LLM-as-a-Judge + Self-Description.

---

**LLM-as-a-Judge + Basic Demographic Information**

# ROLE
You are a meticulous user preference analysis expert. Below I will provide you with this user's basic demographics and this user's preferred responses from their input history.

# Primary Goal
Your primary goal is to understand this user's inherent preferences from the basic demographics, interaction history and use that understanding to judge responses to the user's current input.

# User's basic demographics
[The Start of User's basic demographics]
{basic_demographics}
[The End of User's basic demographics]

# Examples from user's preference history
[The Start of User's preference history]
{few_shots}
[The End of User's preference history]

# INSTRUCTIONS
Follow these steps precisely:

1. Analysis : First, perform your detailed analysis.
a. Analyze User Persona & Preferences: Based on the history, deduce the user's likely persona, communication style, demographic information, and core preferences. Retain only the conjectures of which you are highly confident.
b. Evaluate new Responses against User Persona & Preferences: For each user's input and its corresponding responses, evaluate how well it aligns with the user's persona & preferences you inferred from the history above.
c. Final Decision: State which response you believe is better and briefly summarize the single most important reason.

2.   Final Output (JSON) : After completing your analysis and decision, generate a single, valid JSON object as your final answer. The JSON should be the only thing you output.

# JSON OUTPUT SPECIFICATION
- The JSON object must have exactly two keys: "rationale" and "better_response".
- "rationale": A concise string that explain the user's preference and the advantages of the better response.
- "better_response": An integer, which must be '1' if Response 1 is better, or '2' if Response 2 is better.

Example Output Format:
{{"rationale": "your explanation", "better_response": x}}

# Input Data
[The Start of User Input]\n {user_input}\n [The End of User Input]\n
[The Start of Response 1]\n {response_1}\n [The End of Response 1]\n
[The Start of Response 2]\n {response_2}\n [The End of Response 2]

Figure 14: Prompt of LLM-as-a-Judge + Basic Demographic Information.

**LLM-as-a-judge +Choice Attributes**

# ROLE
You are a meticulous user preference analysis expert. Below I will provide you with examples of this user's preferred responses from their input history, each followed by the user's choice attributes in this conversation, which indicate the rationale for the user's preference for one response over others. A higher score signifies that the corresponding attribute is of greater importance.

# Primary Goal
Your primary goal is to understand this user's inherent preferences from each conversation history together with its corresponding choice attribuets and use that understanding to judge responses to the user's current input.

# Examples from user's preference history and corresponding choice attribuets
[The Start of User's preference history and corresponding choice attribuets]
{few_shots_with_choice_attributes}
[The End of User's preference history and corresponding choice attribuets]

# INSTRUCTIONS
Follow these steps precisely:

1. Analysis : First, perform your detailed analysis.
a. Analyze User Persona & Preferences: Based on the history, deduce the user's likely persona, communication style, demographic information, and core preferences. Retain only the conjectures of which you are highly confident.
b. Evaluate new Responses against User Persona & Preferences: For each user's input and its corresponding responses, evaluate how well it aligns with the user's persona & preferences you inferred from the history above.
c. Final Decision: State which response you believe is better and briefly summarize the single most important reason.

2. Final Output (JSON) : After completing your analysis and decision, generate a single, valid JSON object as your final answer. The JSON should be the only thing you output.

# JSON OUTPUT SPECIFICATION
- The JSON object must have exactly two keys: "rationale" and "better_response".
- "rationale": A concise string that explain the user's preference and the advantages of the better response.
- "better_response": An integer, which must be '1' if Response 1 is better, or '2' if Response 2 is better.

Example Output Format:
{{"rationale": "your explanation", "better_response": x}}

# Input Data
[The Start of User Input]
{user_input}
[The End of User Input]

[The Start of Response 1]
{response_1}
[The End of Response 1]

[The Start of Response 2]
{response_2}
[The End of Response 2]

Figure 15: Prompt of LLM-as-a-judge +Choice Attributes.

