# OpenReview forum: "P-GenRM: Personalized Generative Reward Model with Test-time User-based Scaling"
_ICLR.cc/2026/Conference — ICLR 2026 Oral_

### Official Review · Reviewer_ZALW · 2025-10-23

**Soundness:** 3
**Presentation:** 2
**Contribution:** 3
**Rating:** 4
**Confidence:** 3

**Summary:**

This paper proposes P-GenRM, a personalised generative reward model to address personalised alignment. In more detail, the authors aim to address the challenge of oversimplifying scenario-specific preferences and generalising new users with limited feedback. P-GenRM is a combination of various methods: (i) transforms preference signals into evaluation chains, (ii) clusters users into prototypes, (iii) various stages of training form SFT, to RL, to hard-negative aware curriculum learning.

**Strengths:**

This paper has various strengths, particularly I wish to point out:
- The achieved results on first look are indeed promising, especially Table 1, which makes a strong case for their proposed method
- The experiments, especially the ablation experiments, are extensive and cover most of the questions I had while reading this paper.
- Their overview figures (1 and 2) help to understand what is happening in their method.

**Weaknesses:**

I wish to point out some weaknesses that this paper has, upon which the authors could improve to make a stronger case:

- **missing errorbars**: While Table 1 reports the error bars (I assume this is the standard error?), all other experiments do not report any error bars, which makes it difficult to gauge the statistical significance of the experiments. Especially the results in Table 3 and Figure 3b could have overlapping errors.

- **Composition of many methods**: While I appreciate the work of the authors in explaining and displaying the various parts of their methods, to me (personally), the final method seems a bit over-engineered. This makes it difficult to gauge which part has a more substantial effect on good performance, and I also found it hard for the reader to follow. Table 3 does a good job in slightly mitigating this problem

- **What about Generation?** To me, the reason to have a reward model is to train a policy model which demonstrates personalised alignment characteristics subsequently. While I understand the authors' focus on the accuracy of the reward model, it would also be beneficial to test whether this RM can be used to train a policy model and whether personalisation still works.

- **Is average accuracy the right metric?** If I understand it correctly, the authors use average accuracy as a metric in most tables. This would make sense if we are interested in general preference learning. However, as we are looking into personalised preference learning, it is essential to investigate accuracy across the different types of personas in the dataset. Because if there is a majority class/persona in the dataset, and the RM learns its preferences, the average accuracy will look good, while not actually achieving personalised alignment.

**Misc**
- I did not find any code available to inspect what is happening
- The authors refer to the LLM-as-a-judge output as process reward, despite only giving a single value in return. I would abstain from using this terminology, as process reward models (at least from my understanding) are (semi-) dense reward models that assign intermediate rewards to tokens, rather than just one value at the end.

**Questions:**

- How would you implement P-GenRM for generation in a policy model?
- Could this potentially even be extended to Direct Preference Optimisation (DPO) as an implicit reward model?

---

> ### Author Response · Authors · 2025-11-23
> **Response to Reviewer ZALW (Part 1/2)**
>
> Dear Reviewer,
>
> We sincerely appreciate your constructive suggestions and your recognition of our strenghths. We apologize for any ambiguities in the original presentation that may have hindered a clear understanding of our work. To address your concerns, we have conducted extensive additional experiments and revised the manuscript accordingly. Below, we provide our point‑by‑point responses with further explanations.
>
> > **W1: Missing errorbars**
>
> We thank you for your rigorous suggestion. In the initial version of the paper, we omitted the error bars *solely for visual clarity in table and figure*. We have now added the error bars in the manuscript.
>
> > **W2: Composition of many methods**
>
> We apologize for any confusion caused by our original writing and have revised the manuscript accordingly.
>
> Although P-GenRM is composed of several components, each part serves a distinct and necessary purpose, and Table 3 shows that removing any component leads to consistent and significant performance degradation. Below, we provide a concise explanation of the motivation and contribution of each component.
>
> During training, P-GenRM undergoes a three-stage process:
>
> In the *first-stage SFT*, we distill evaluation chains from o3 to enable the base model (LLaMA 3.1 8B-Instruct) to acquire the ability to perform personalized analysis, including user persona generation, preference inference, and personalized scoring. SFT brings clear improvements, but the gap with o3 remains significant **(Chatbot Arena: 66.77 vs 69.14; Prism: 57.08 vs 63.87**, the former represents P-GenRM-8B **)**.
>
> The *second-stage reinforcement learning*  serves two key purposes: (1) it enables the model to move beyond merely imitating the O3 rating patterns and to explore more optimal personalized scoring strategies; and (2) it improves the robustness of P-GenRM's evaluation chain generation, particularly in scenarios where explicit preferences are missing.  P-GenRM sees substantial performance gains after this stage  **(Chatbot Arena: 66.77 → 71.07; Prism: 57.08 → 63.82)**.
>
> The purpose of the *third-stage curriculum learning* is to encourage thorough exploration on hard negative samples (e.g., those that even o3 consistently mislabels), thereby enhancing the model’s ability to handle challenging cases.  After the three-stage training, P-GenRM moves beyond analytical imitation of O3 and successfully discovers effective personalized scoring strategies. As a result, it not only outperforms O3 but also achieves state-of-the-art performance  **(Chatbot Arena: 72.68 vs 69.14; Prism: 65.32 vs 63.87**, the former represents P-GenRM-8B **)**.
>
> The proposed *Test-time User-based Scaling* is designed to fully exploit the inherent test-time scalability of GenRM, with two main objectives:
>
> (1) **Enhancing the reliability of preference inference**. Here, *individual-level scaling* can be viewed as exploring multiple hypotheses about a user’s preferences to obtain richer and more robust scoring rubrics and thereby improves inference reliability. *Prototype-level scaling*, on the other hand, corrects potential errors in current preference inference by incorporating preferences from similar users.
>  (2) **Improving generalization to new users**, primarily by assigning a new user to an appropriate user prototype and simulating their preferences through those of similar users.
>
> This design leads to a further notable improvement for P-GenRM  **(Chatbot Arena: 72.68 → 75.92; Prism: 65.32 →  68.06)**, moreover, bringing its performance on an OOD dataset on par with Qwen-2353B-A22B.
>
> > **W3: What about Generation? & Q1: How to implement P-GenRM for generation in a policy model?**
> >
> > **Q2 P-GenRM as the implicit reward model in DPO**
>
>  We agree that assessing the downstream policy model is essential for further validating the effectiveness of the reward model and **have added this experiment in Section 5.5 and Appendix A.14**. In response, we have conducted extensive experiments using both GRPO and DPO to train policy models with P-GenRM and evaluate their personalization performance. The details are as follows:
>
> We train the policy model on Llama 3.1 8B-Instruct using the Chatbot Arena–Personalized dataset under two different setups.:  **(1) P-GenRM is integrated into GRPO to score each response and compute the corresponding relative advantage**. Specifically, given the user’s historical preference pairs, P-GenRM assesses how well each candidate response aligns with the user’s preferences (using the same prompting format as in P-GenRM's training). These scores are then incorporated into advantage estimation and loss computation to update the policy model.
>
> **(2) P-GenRM serves as the implicit reward model in DPO.** Given a pair of candidate responses, the one assigned a higher score by P-GenRM is treated as *chosen*, while the lower-scored one is treated as *rejected*. These preference pairs are then used to train the policy model via DPO.

---

> ### Author Response · Authors · 2025-11-23
> **Response to Reviewer ZALW (2/2)**
>
> The policy model is evaluated as follows: given a user’s historical preferences, we prompt the model to generate responses that align with the user’s tastes. We then employ three advanced LLMs (GPT-4o, Claude-sonnet-4, Gemini 2.5-Pro) as judges to rate the personalized quality of each response on a *1–5* scale. For each query, we record the average score across the three judges, and report the overall results below.
>
> | Policy Model              | GPT-4o  $\quad$| Claude-sonnet-4 $\quad$| Gemini-2.5-pro $\quad$| AVG Score |
> | ------------------------- | ------ | --------------- | -------------- | --------- |
> | Llama3.1-8B-Instruct      | 3.30   | 2.96            | 2.96           | 2.95      |
> | Qwen2.5-7B-Instruct       | 3.16   | 2.92            | 2.77           | 2.98      |
> | Llama3.1-70B-Instruct     | 3.33   | 3.05            | 3.08           | 3.16      |
> | Qwen2.5-72B-Instruct      | 3.42   | 3.14            | 3.17           | 3.24      |
> | Llama3.1-8B-Instruct-DPO  | 3.49   | 3.15            | 3.28           | **3.31**  |
> | Llama3.1-8B-Instruct-GRPO | 3.47   | 3.22            | 3.39           | **3.35**  |
>
> **The *8B* policy models trained with P-GenRM under DPO/GRPO setting exhibit substantial improvements over their corresponding base models, outperforming even larger (*70B*) instruct models**. These results provide strong evidence for the effectiveness of P-GenRM in policy model training.
>
>
>
> > **W4: Is average  accuracy the right metric?**
>
> We thank the reviewer for this constructive comment and **have included this information in Section Experiments and Appendix A.6**. Since prior work [1–4] commonly adopted accuracy as the primary evaluation metric, we initially followed this setting for consistency. However, we agree that in the context of *personalized preference learning*, evaluating performance across user groups of different sizes provides a more faithful measure of personalization capability.
>
> To address this concern, we conducted an additional prototype-wise analysis using **macro accuracy**, where accuracy is computed separately for each persona group and then averaged across all groups, ensuring that minority personas are equally weighted. Based on our clustering method, the user groups in the PRISM dataset exhibit a clear long-tail distribution:
>
> |                | Min  | Max  | Mean  | Median | 25th pct | 75th pct |
> | -------------- | ---- | ---- | ----- | ------ | -------- | -------- |
> | Num of samples | 18   | 930  | 154.8 | 105    | 63       | 176      |
>
> **Despite this pronounced long-tail distribution**, our method maintains stable performance across prototypes: The prototype-level macro accuracy is **0.6521**, and the sample-level average accuracy is **0.6532**, differing by only **0.0011**, indicating that the model does not overfit to majority personas. Moreover, persona-level accuracies have moderate dispersion (median **0.6500**, std **0.0401**, IQR **0.0544**), showing that the model performs consistently across both large and small persona groups. Detailed results for P-GenRM-8B across all 50 prototypes are provided in the **Appendix A.6**.
>
> |                     | Macro  | Min    | Max    | Median | Std    | IQR    |
> | ------------------- | ------ | ------ | ------ | ------ | ------ | ------ |
> | Prototype-level Acc | 0.6521 | 0.5806 | 0.7508 | 0.6500 | 0.0401 | 0.0544 |
>
> We further compare the **macro accuracy** performance of P-GenRM against other baseline methods:
>
> |                    | Llama3.1-8B   | Llama3.1-70B  | o3            | Fine-tuned BT-70B | SynthesizeMe 70B | P-GenRM-8B         |
> | ------------------ | ------------- | ------------- | ------------- | ----------------- | ---------------- | ------------------ |
> | **macro accuracy** | 56.24 ± 1.74% | 58.27 ± 1.71% | 63.33 ± 1.52% | 60.64 ± 1.87%     | 61.51 ± 1.63%    | **65.21** ± 1.42% |
>
> **P-GenRM achieves the highest macro accuracy (65.21%) among all evaluated baselines**, outperforming both open-source and proprietary models with strong prompting strategies.
>
> [1]. SynthesizeMe! Inducing Persona-Guided Prompts for Personalized Reward Models in LLMs. ACL 2025 Main
>
> [2]. Personalizing Reinforcement Learning from Human Feedback with Variational Preference Learning. Neurips 2024
>
> [3]. Group Preference Optimization: Few-Shot Alignment of Large Language Models. ICLR 2024
>
> [4]. PAL: Pluralistic alignment framework for learning from heterogeneous preferences. NeurIPS 2024 Workshop
>
> > **Misuse of  the term 'process reward'**
>
> We thank the reviewer for this rigorous advice. We will instead use the term “reasoning quality reward,” which directly reflects that our purpose is to assess the evaluation  chain's quality
>
> **We would like to thank you again for your constructive review**, if any of our response needs further clarification, please inform us and we'll promptly follow up.

---

> > ### Comment · Reviewer_ZALW · 2025-11-25
> >
> > I wish to thank the authors for all their work done in the rebuttal. I will address each response here separately.
> >
> > *A1) errorbars*
> > Thank you for introducing the error bars (I assume these are the standard errors; is that right?). After inspecting Tables 2 and 3, it is now clear that there is significant overlap in the standard errors (especially in Table 2, Chatbot Arena in Table 3, and Table 4). I therefore take from this that the results are not really sufficiently robust to support the authors' claims. At least the mean looks good, but that is the point of the error bars: to highlight this weakness.
> >
> > *A2) clarifying the method*
> > Thank you for clarifying the method. It is acknowledged and appreciated.
> >
> > *A3) Generation*
> > Thank you for the follow-up experiment; it is appreciated, and it actually seems to have a positive effect. Any chance to add the error bars there too, to determine if this is actually significant, before claiming that an 8B model outperforms a 70B model?
> >
> > *A4 Macro average*
> > Thank you for the follow-up experiment! This seems convincing.
> >
> > I am willing to raise my score if my concerns in A3 can be addressed.

---

> ### Author Response · Authors · 2025-11-26
> **Further clarification (1/2)**
>
> Dear Reviewer ZALW:
>
> We genuinely thank the reviewer again for your valuable time and constructive comments. Following the reviewer’s suggestion, we address A3 first with **additional experiments and  statistical analyses** to resolve the major remaining concern. We then respond to A1 with clarifications and revisions to ensure that the error bars and corresponding claims are accurately presented.
>
> > **A3: Generation results with error bars**
>
> We thank the reviewer for this thoughtful advice. In response, we repeated **5** independent runs of the evaluation.   For each run, the policy model generated its responses for the full test set, and we recorded the overall scores across the three evaluators (GPT-4o, Claude-Sonnet-4, and Gemini-2.5-Pro).  We then computed the **mean,** **standard error (SE),** and **95% confidence intervals (CI)**  over these **5** runs. The results are as follows, and we provide full per-run results in **Appendix A.14** for transparency.
>
> | Policy Model              | Mean  | SE     | 95% CI          |
> | ------------------------- | ----- | ------ | --------------- |
> | Llama3.1-8B-Instruct      | 2.954$\quad$ | 0.0074 $\quad$| [2.939 , 2.969] |
> | Qwen2.5-7B-Instruct       | 2.970 $\quad$| 0.0089$\quad$ | [2.952 , 2.988] |
> | Llama3.1-70B-Instruct     | 3.156$\quad$ | 0.0093$\quad$ | [3.138 , 3.174] |
> | Qwen2.5-72B-Instruct      | 3.214 $\quad$ | 0.0089 $\quad$| [3.192 , 3.228] |
> | **Llama3.1-8B-Instruct-DPO** $\quad$ | 3.316 $\quad$| 0.0068 $\quad$| **[3.303 , 3.329]** |
> | **Llama3.1-8B-Instruct-GRPO** $\quad$ | 3.354 $\quad$| 0.0102 $\quad$|**[3.334 , 3.374]** |
>
> Across five independent runs, the 8B policy models trained with P-GenRM achieved 95% confidence intervals whose **lower bounds** are **3.303** and **3.334**, both of which are **higher** than the **upper bounds** of the 70B-sized models (3.174 and 3.228), showing **no interval overlap**.  This confirms that the performance advantage is robust and statiscally significant. We **have updated Appendix A.14** accordingly to include these results.
>
>
> > **A1: Error bars in the manuscript**
>
> We thank the reviewer for raising this important point.  We apologize for this unclarity that the error bars presented in our previous revision are based on **standard deviations** over 5 independent runs. We now report the error bars in **standard errors (SE)**  in line with  the reviewer’s suggestion. The updated tables are as follows:
>
> **Table 2: Ablation studies**
>
> | Method          | Chatbot Arena (Mean ± SE) | PRISM (Mean ± SE) |
> | --------------- | ------------------------- | ----------------- |
> | P-GenRM         | 72.68 ± 1.85              | 65.32 ± 0.56      |
> | w/o CL          | 71.07 ± 1.44              | 63.82 ± 0.64      |
> | w/o CL, PR      | 70.22 ± 1.74              | 62.70 ± 0.73      |
> | w/o CL, OR      | 69.05 ± 1.59              | 60.94 ± 0.77      |
> | w/o CL, RL      | 66.76 ± 1.42              | 57.08 ± 0.89      |
> | w/o CL, RL, SFT | 56.37 ± 2.18              | 52.04 ± 0.54      |
>
> **Table 3: Comparison of adaptive and static persona**
>
> | Method                | Chatbot Arena (Mean ± SE) | PRISM (Mean ± SE) |
> | --------------------- | ------------------------- | ----------------- |
> | Qwen3–8B              | 61.82 ± 1.47              | 55.01 ± 0.77      |
> | Qwen3–8B + SMe        | 62.57 ± 1.84              | 56.33 ± 0.91      |
> | Qwen3–8B + PSI (ours) | 64.22 ± 1.58              | 58.01 ± 0.83      |
> | o3                    | 64.47 ± 1.62              | 56.34 ± 0.64      |
> | o3 + SMe              | 67.73 ± 1.94              | 58.49 ± 1.22      |
> | o3 + PSI (ours)       | 69.14 ± 1.46              | 63.87 ± 0.85      |
>
> **Table 4: P-GenRM's performance  at different scaling scales**
>
> | Model             | Chatbot Arena (Mean ± SE) | PRISM (Mean ± SE) |
> | ----------------- | ------------------------- | ----------------- |
> | o3                | 64.47 ± 1.62              | 56.34 ± 0.64      |
> | o3 + PSI          | 69.14 ± 1.46              | 63.87 ± 0.85      |
> | P–GenRM (8B)      | 72.68 ± 1.85              | 65.32 ± 0.56      |
> | + Ind-8           | 73.61 ± 1.54              | 65.79 ± 0.68      |
> | + Ind-4 , Pro-4   | 73.66 ± 1.39              | 66.20 ± 0.75      |
> | + Ind-16          | 73.87 ± 1.69              | 66.66 ± 0.82      |
> | + Ind-8 , Pro-4   | 74.30 ± 1.60              | 67.54 ± 0.58      |
> | + Ind-8 , Pro-8   | 74.89 ± 1.75              | 67.44 ± 0.84      |
> | + Ind-32          | 75.59 ± 1.64              | 67.65 ± 0.66      |
> | + Ind-16 , Pro-8  | 75.92 ± 1.70              | 68.06 ± 0.69      |
> | + Ind-0 , Pro-8   | 66.90 ± 1.54              | 57.65 ± 0.86      |
> | + Ind-16 , Pro-16 | 72.59 ± 1.61              | 64.61 ± 0.72      |

---

> ### Author Response · Authors · 2025-11-26
> **Further clarification (2/2)**
>
> Based on the standard errors, some degree of overlap remains, primarily on the Chatbot Arena–Personalized dataset. This is likely due to its substantially smaller size compared with PRISM (*1,170* unique queries vs. *10,935*), which results in higher variance for both the baselines and our method.
>
> Nevertheless, incorporating each key component yields consistent and meaningful improvements across both datasets, with only modest or no overlap in the corresponding error bars. For example:
>
>  The progressive inclusion of SFT, RL, and CL in Table 2 leads to steadily higher performance (56.37 ± 2.18 → 66.76 ± 1.42 → 71.07 ± 1.44 → 72.68 ± 1.85).
>
> Likewise, the adaptive persona mechanism in Table 3 offers clear advantages  (63.87 ± 0.85 vs. 58.49 ± 1.22; 69.14 ± 1.46 vs. 67.73 ± 1.94).
>
> The comparison with and without test-time user-based scaling further shows meaningful gains (75.92 ± 1.70 vs. 72.68 ± 1.85; 68.06 ± 0.69 vs. 65.32 ± 0.56).
>
> *Taken together, these results support the soundness of our design choices.*
>
> Furthermore, we conduct Welch’s t-tests comparing our best performances with the previous SOTA methods on PersonalRewardBench. The results are as follows:
>
> | **Welch’s t-test**$\quad$ | **Chatbot Arena-personalized** $\quad$| **PRISM**      |
> | ------------------ | ------------------------------ | -------------- |
> | p-value            | 0.047 < 0.05                   | 0.0009 < 0.001 |
>
> *This demonstrates that our improvements over previous works are statistically significant.*
>
> We apologize again for not explaining the interpretation of the error bars more clearly in earlier revision. In future work, we will investigate approaches for reducing variance under limited user data.
>
> **We deeply thank the reviewer again for your valuable responses and constructive suggestions**, which allow us to improve the quality of our work. If any further clarification is needed, please inform us and we will promptly provide additional information.

---

> > ### Comment · Reviewer_ZALW · 2025-11-27
> >
> > Thank you for this very detailed and rigorous follow-up.
> >
> > I apologise for being so insistent on the error bars. This analysis significantly improves the manuscript, in my opinion. I am raising my score to 6.

---

> > > ### Author Response · Authors · 2025-11-27
> > > **Sincere Appreciation for Your Review**
> > >
> > > Dear Reviewer ZALW:
> > >
> > > We would like to express our sincere gratitude once again for your constructive suggestions and thoughtful comments, which enabled us to improve the quality of our work. We deeply appreciate your support and endorsement!
> > >
> > > Sincerely,
> > >
> > > All authors

---

### Official Review · Reviewer_qy8U · 2025-10-29

**Soundness:** 2
**Presentation:** 2
**Contribution:** 2
**Rating:** 4
**Confidence:** 3

**Summary:**

This paper introduces P-GenRM, a personalized generative reward model that aligns large language models to individual user preferences. This is by generating structured evaluation chains, clustering users into prototypes, and applying a dual-granularity scaling mechanism that adapts scoring both at the individual and group levels. This approach reduces noise, improves generalization to new users, and achieves good performance with additional gains from test-time user-based scaling.

**Strengths:**

1. This paper address an important problem of user personalization by providing the full pipeline of collecting data, clustering users, refining the personalized reward model and adapts the output.
2. The experimental evaluation is comprehensive.
3. The pipeline uses both implicit and explicit preference signals, which fully utilizes the preference dataset.

**Weaknesses:**

1. The main concern is the limited novelty of the paper. It seems that the main contribution of this paper is proposing the overall pipeline of obtaining personalized outputs, by using existing methods such as generative reward models and clustering users. It is not very clear what the technical contributions are.

2. Lack of analysis of inference costs. It would be nice if some analysis on the costs of personalization can be done, including analysis of the baselines.

**Questions:**

What are the practical limitations of this method? I suggest including a limitation section in the paper.

---

> ### Author Response · Authors · 2025-11-23
> **Response to Reviewer qy8U (Part 1/2)**
>
> Dear Reviewer,
>
> we sincerely appreciate your important suggestions and your recognition of our paper’s strengths. We apologize for any lack of clarity in our original writing. In response, we have added additional experiments and  below is our point-by-point response  trying to addressing  all of your concerns and clarify our technical contributions.
>
> > **W1: limited novelty and unclear technical contributions**
>
> We thank the reviewer for raising this important point. We provide a clarification of our contributions as follows:
>
> While our approach indeed uses existing components such as generative reward models and user clustering, the problem we address in personalized alignment remains significantly under-explored:
>
> > **Given the broad and heterogeneous preference space created by diverse users and contexts, how can we accurately infer a specific user’s scenario-specific preferences and produce reliable personalized judgments at test time?**
>
>  Our contributions lie in providing a *concrete, end-to-end solution* to this challenge. Specifically,
>
> **1.** We advance **adaptive (online) preference modeling** by transforming hybrid interaction signals into scenario-specific personas and scoring rubrics,  overcoming the limitation of prior works that treat preferences as a fixed set of principles, making them unable to adapt to different users or contexts.
>
> **2.** We introduce a novel test-time user-based scaling mechanism. **It simultaneously enhances the reliability of preference inference and improves generalization to new users**: at the individual level, it explores multiple hypotheses about a user’s preferences to obtain richer and more robust scoring rubrics, while at the prototype level it refines these inferences using preferences from similar users. For new users under sparse historical data, assigning them to suitable prototypes allows the model to approximate their preferences through those of similar users.
>
> **3.** We provide a complete three-stage training framework for enabling  reward models to deliver accurate personalized reward signals in open-ended dialogue settings. Combined with our test-time user-based scaling mechanism, the resulting P-GenRM-8B achieves substantial gains over  advanced proprietary LLMs (e.g., o3) and prior SOTA methods by an average of approximately **3.97%**.
>
> Given that we are the first to propose a comprehensive framework for adaptive preference modeling together with a new test-time scaling paradigm, and considering the promising empirical results our approach achieves, we believe that our work makes a substantial contribution to the community.
>
> > **W2: Lack of analysis of inference costs**
>
> We thank the reviewer for this important suggestion and **have included this analysis  in Section Experiment and  Appendix A.10**. We decompose the total inference cost   into three parts: **(1) Prompt encoding (KV-cache construction):** The inclusion of historical user preference selections results in a long input prompt, making KV-cache construction the dominant component of inference latency. This cost is incurred only once per query. **(2) Response generation:** Decoding is lighter than prompt encoding and supports parallelism, contributing only a small portion of total cost.  **(3) Test-time User-based Scaling:** The module  yields substantial performance gains while incurring limited addtional computational cost:
>
> For **individual-level scaling**, we perform parallel sampling via the OpenAI-compatible `n` parameter, allowing multiple outputs to be generated in a single model call with latency comparable to single-output generation; For **prototype-level scaling**, similarity computation is lightweight and  the preference-based scoring over similar users can also be parallelized efficiently handled through vLLM’s batching capabilities.
>
> We measured the overall inference latency of our method and baselines on the full Chatbot Arena-Personalized test set. All open-source models and our method were deployed using **vLLM** on **8 NVIDIA A100 GPUs**, and proprietary models were accessed via the Alibaba Cloud API (maximum concurrency: 40).

---

> ### Author Response · Authors · 2025-11-23
> **Response to   Reviewer qy8U (Part 2/2)**
>
> The comparative results are as follows:
>
> | Model              $\quad$             | Inference Time (Wall-clock) $\quad$ | Performance |
> |---------------------------------|------------------------------|-------------|
> | LLaMA3.1-8B-Instruct + PSI    $\quad$     | 00:14:06                     | 62.20       |
> | LLaMA3.1-70B-Instruct + PSI    $\quad$   | 00:39:17                     | 65.55   |
> | SynthesizeMe + FT RM-8B     $\quad$      | 00:24:10                     | 69.78       |
> | SynthesizeMe + FT RM-70B   $\quad$       | 01:29:59                     | 72.05       |
> | o3+PSI                $\quad$            | 01:25:45                     | 69.14       |
> | P-GenRM-8B        $\quad$                | **00:14:16**                     | **72.68**       |
> | P-GenRM-8B + Ind-8, Pro-4    $\quad$     | **00:18:22**                     | **74.30**       |
> | P-GenRM-8B + Ind-16, Pro-8    $\quad$    | **00:23:05**                     | **75.92**   |
>
> P-GenRM-8B achieves the best performance (up to 75.92 vs 72.05) with **lower** inference time than baseline methods. The proposed Test-time User-based Scaling further yields substantial performance gains (**3.24%**) while introducing only limited additional inference cost (**00:14:16 → 00:23:05**), demonstrating an effective trade-off between latency and quality.
>
>
>
> > **Q1: Practical limitations of this method**
>
> Despite strong empirical performance, we acknowledge  two current limitations of the method: (1). It requires generating an evaluation chain to obtain reliable personalized scores, which may be less efficient than reward models that directly produce scalar values  when considering inference speed alone; (2). It relies on three historical  preference selections to construct reasonable preference analysis, which implies a moderate amount of data collection in practical scenarios.
>
> **We would like to thank you again for your constructive review**, if any of our response needs further clarification, please inform us and we'll promptly follow up.

---

> > ### Comment · Reviewer_qy8U · 2025-11-25
> >
> > Thank you for the clarification, my concerns are mostly addressed and I am raising my score to 6.

---

> > > ### Author Response · Authors · 2025-11-26
> > > **Sincere Appreciation for Your Review**
> > >
> > > Dear Reviewer qy8U:
> > >
> > > We would like to express our gratitude once again for your valuable time and constructive suggestions. Thank you very much for approving our work!
> > >
> > > Sincerely,
> > >
> > > All authors

---

### Official Review · Reviewer_Ehiq · 2025-10-31

**Soundness:** 3
**Presentation:** 2
**Contribution:** 2
**Rating:** 6
**Confidence:** 2

**Summary:**

The paper tackles personalized preference modeling for LLM outputs with a generative reward model, P-GenRM, tailored to each user. The method trains a model that outputs a short persona and a weighted scoring rubric from a user’s history and any stated preferences, enabling clearer, user-aware judgments. The paper proposes a complex three-step training recipe to train their reward model that supports test-time scaling that averages multiple runs for the same user and also brings in signals from similar users to cut noise and handle cold-start. The experiments validate the approach on personalized benchmarks with consistent gains over prior methods.

**Strengths:**

The paper is generally a solid paper. The strengths include:

1. The motivation is clear and important: personalized preference modeling is a real bottleneck for aligning LLM outputs to individual users.

2. Conceptually, the method advances online preference handling by turning user evidence into a contextual persona and rubric, and by scaling judgments at test time with both multiple runs for the same user and signals from similar users; this is a novel and well-argued design.

3. The approach integrates several effective components—supervised imitation, RL with process and outcome signals, hard-negative curriculum, and prototype-based test-time scaling—and the ablations substantiate that each piece contributes.

**Weaknesses:**

There are still some weaknesses listed as below.

1. The method in the main text is too abstract. Key I/O and losses are not clearly written there or clearly pointed to the appendix. The algorithm, including both training and test-time, is generally complex, and I have some questions to be answered in the question section.

2. Computation cost is underreported. I assume the additional test-time user-based scaling may take much more than the baselines. A computation cost ablation study may be necessary.

3. Minor writing/format issues:
   1. Capitalization after a comma: “Based on this, The personalized generative reward model…”
   2. In table 3, "w/o Cl, RL, SFT" should be w/o CL, RL, SFT

**Questions:**

1. The paper proposes a three-stage training process. I think the training of generative reward models has been studied in previous works, as mentioned in the related works. What is the relation between your training procedure and previous works exactly? Why do you do so intuitively?
2. For the experimental parts, I have two problems. a) To use your test-time scaling, what is the computation cost to have your results compared to the baselines? b) How many samples are required to generate a reasonable preference for each user?

---

> ### Author Response · Authors · 2025-11-23
> **Response to Reviewer Ehiq (Part 1/2)**
>
> Dear Reviewer,
> we sincerely appreciate your constructive suggestions and your recognition of our paper’s strengths. We apologize for any lack of clarity in our original writing. In response, we have added additional experiments and revised the manuscript accordingly. Below, we provide a point-by-point response  trying to addressing  all of your concerns.
>
> > **W1: Unclear specification of key I/O and loss functions in the training and testing procedures**
>
> We acknowledge this unclarity in our original presentation and **have modfied Section Methodology** accordingly.
> Below, we clarify this ambiguity with more explanations. Let $ H_t^{(u)} $ denote the historical preference choices of user $u$, $ E^{(u)} $ the explicit preference criteria (if provided), $ q_t $ the current input, and $ y_t^i $ the i-th  candidate responses. Each training stage (SFT, RL, curriculum learning)  and inference takes $ q_t $, $ H_t^{(u)} $, and $ y_t^i $ as input and outputs the structured evaluation chain $[P_t^{(u)}; S_t^{(u)}]$, where $P_t^{(u)}$ is a textual preference analysis for the current user and context and $S_t^{(u)}$  is a weighted set of scoring rubrics and the corresponding scores assigned to the candidate responses, only SFT may additionally use explicit preference signals $ E^{(u)} $.
>  We summarize the outputs of the three-stage training and testing  in the table below:
> |                  | **First Stage SFT**                                   | **Second Stage RL**  $\quad$                  | **Curriculum Learning**       $\quad$      | **Inference**                            |
> |------------------|--------------------------------------------------------|------------------------------------------|------------------------------------------|-------------------------------------------|
> | **Input**        | $q_t, H_t^{(u)}, E^{(u)},\ y_t^i \quad$  | $q_t, H_t^{(u)}, y_t^i $                 | $q_t, H_t^{(u)}, y_t^i$                 | $q_t, H_t^{(u)}, y_t^i$                  |
> | **Output**       | $[P_t^{(u)};\ S_t^{(u)}]$                              | $[P_t^{(u)};\ S_t^{(u)}]$               | $[P_t^{(u)};\ S_t^{(u)}]$               | $[P_t^{(u)};\ S_t^{(u)}]$                |
>
> We also provide the  detailed objective function of GRPO in  RL stage and curriculum learning in Section 4.1, Line 227.
>
> > **W2 and Q2 a): The compuation cost caused by the test-time user-based scaling**
>
> We thank the reviewer for this important suggestion and **have included this analysis  in Section Experiment and  Appendix A.10.**  We measured the end-to-end inference time of P-GenRM-8B at different scaling levels on the full Chatbot Arena-Personalized test set and compared it against several baselines. Both the open-source models and our method were deployed using **vLLM** on **8 NVIDIA A100 GPUs**, while proprietary models were accessed via the Alibaba Cloud API with a maximum concurrency of 40. The results are as follows:
> | Model                           | Inference Time (Wall-clock) | Performance |
> |---------------------------------|------------------------------|-------------|
> | LLaMA3.1-8B-Instruct + PSI      | 00:14:06                     | 62.20       |
> | LLaMA3.1-70B-Instruct + PSI     | 00:39:17                     | 65.55   |
> | SynthesizeMe + FT RM-8B         | 00:24:10                     | 69.78       |
> | SynthesizeMe + FT RM-70B        | 01:29:59                     | 72.05       |
> | o3+PSI                          | 01:25:45                     | 69.14       |
> | P-GenRM-8B                      | **00:14:16**                     | **72.68**       |
> | P-GenRM-8B + Ind-8, Pro-4       | **00:18:22**                     | **74.30**       |
> | P-GenRM-8B + Ind-16, Pro-8      | **00:23:05**                     | **75.92**   |
>
> **The observed increase in inference time for our method remains limited (00:14:16 → 00:23:05), and it outperforms larger models while requiring less inference time.** We believe this is mainly due to two factors:
>
> **First**, the task requires including a certain number (specifically, 3) of prior user preference selections in the prompt to allow the model to infer meaningful user preferences. This leads to long input sequences, making the construction of the KV cache for the prompt the major component of inference latency. This prompt encoding is performed exactly once per query and shared across all samples.
>
> **Second**, our test-time user-based scaling introduces only limited additional cost. It consists of two parts:
>  – For **individual-level scaling**, we perform parallel sampling via the OpenAI-compatible `n` parameter, allowing multiple outputs to be generated in a single model call with latency comparable to single-output generation.
>  – For **prototype-level scaling**, similarity computation is lightweight and  the preference-based scoring over similar users can also be parallelized efficiently handled through vLLM’s batching capabilities.

---

> ### Author Response · Authors · 2025-11-23
> **Response to Reviewer Ehiq (Part 2/2)**
>
> As a result, the overall increase in inference cost remains modest, especially considering the performance improvements brought by the Test-time User-based Scaling, which yields a **3.24%** gain over P-GenRM-8B without scaling and a **3.87%** advantage with less inference time  over the previous state-of-the-art .
>
> > **W3: Minor writing/format issues**
>
> We thank the reviewer for this helpful suggestion. We have corrected these errors and carefully reviewed the rest of the manuscript to eliminate any remaining typos.
>
>
>
> > **Q1: Intuition and relation to prior works of the proposed three-stage training procedure**
>
> We thank the reviewer for this insgihful comment. Our approach generally follows the *SFT → RL* paradigm similar to prior works [1, 2, 3], where SFT equips the model with basic analytical abilities and RL further enhances the quality of its reasoning. Building on this foundation, **we introduce three key extensions to address the challenges specific to personalized preference alignment**.
>
> **First**, we  show that user persona are effective priors for personalized scoring, which determines the composition of the structured evaluation chain trained through SFT.
>
> **Second**, since explicit preference signals are typically unavailable at inference time, we prompt the model to infer missing criteria during RL training, in order to improve the robustness of high-quality evaluation chain generation.
>
> **Third**, deriving accurate user preferences from historical behavior is inherently subjective and remains underexplored. Even state-of-the-art models such as o3 struggle with hard cases. To address this, we add a curriculum learning phase after RL, allowing the model to thoroughly explore these hard negative samples.
>
> Together, the three-stage training enables P-GenRM to generate high-quality personalized evaluation chains, and  our ablations in Table 3 confirm that  removing any component leads to consistent and significant performance degradation.
>
>
>
> [1] Inference-Time Scaling for Generalist Reward Modeling  	arXiv:2504.02495
>
> [2] StepWiser: Stepwise Generative Judges for Wiser Reasoning   arXiv:2508.19229
>
> [3] RM-R1: Reward Modeling as Reasoning.  arXiv:2505.02387
>
>
>
> > **Q2 b): Number of samples required for generating reasonable user preference**
>
> We thank the reviewer for pointing out this important detail and **have included this experiment in Appendix A.12**. All the results reported in our orginal paper are based on each user having at least **3** historical preference pairs. Intuitively, a single preference instance only reflects a one-off choice, and two instances are still insufficient to form a consistent pattern. In contrast, three preference samples provide the minimal structure needed to assess preference consistency, reduce randomness, and support reliable personalization by the model. We also conducted experiments on Chatbot Arena-Personalized evaluating P‑GenRM‑8B trained with 1–4 preference samples per user, and the performance differences are shown below:
>
> | Preference Pairs | $\quad$   $\quad$  1            |    $\quad$  $\quad$2            |   $\quad$  $\quad$ 3            |  $\quad$$\quad$   4            |
> | ---------------- | ------------ | ------------ | ------------ | ------------ |
> | Accuracy (%)     | 59.78 ± 5.94 $\quad$| 64.62 ± 4.63 $\quad$| 72.68 ± 4.14 $\quad$| 72.50 ± 3.67 |
>
> We observe that the model performance improves significantly when the number of preference samples reaches **3**. Adding more samples beyond this point primarily contributes to improved evaluation stability, but does not necessarily lead to substantial performance gains. Moreover, we hope to minimize the extent to which the model’s performance relies on early-stage data collection. Therefore, we set the number of preference samples to **3** for both training and evaluation.
>
> **We would like to thank you again for your constructive review**, if any of our response needs further clarification, please inform us and we'll promptly follow up.

---

> > ### Comment · Reviewer_Ehiq · 2025-11-27
> >
> > Thanks to your comprehensive response. My concerns are mainly addressed. I increase my confidence score, but keep the rating.

---

> > > ### Author Response · Authors · 2025-11-27
> > > **Sincere Appreciation for Your Review**
> > >
> > > Dear Reviewer Ehiq:
> > >
> > > We thank the reviewer once again for your valuable time and constructive feedback, which enabled us to  improve the quality of our work.  We deeply appreciate your support and endorsement!
> > >
> > > Sincerely,
> > >
> > > All authors

---

### Author Response · Authors · 2025-12-02
**Summary Information**

Dear AC, SAC and PC,

 Thanks very much for your kind attention. We also sincerely thank the reviewers for their careful evaluation of our work and for recognizing our efforts in previous discussion phase. All reviewers confirmed their concerns were addressed, resulting in final scores of **6, 6, 6**. To facilitate your evaluation, we summarize the key information  below.

**Summary of Our Work**

P-GenRM is the first personalized generative reward model that infers accurate, user-specific rewards in open-ended scenarios. We propose a comprehensive framework for adaptive preference modeling together with a new test-time scaling paradigm. The key idea is to transform diverse preference signals into structured evaluation chains that derive contextual personas and scoring rubrics, and to further scale judgments at test time using both individual and prototype-level preferences. This yields the following strength:

- **Advanced Preference Modeling with Strong Performance and Generalization**: P-GenRM effectively adapts to diverse users and scenarios. Using only an 8B model, it achieves a **3.97%** improvement over prior SOTA methods and demonstrates generalization performance surpassing Qwen-235B-A22B.

- **Test-time Computational efficiency**: Our Test-time User-based scaling mechanism yields a further **3.24%** improvement with limited inference overhead, and it outperforms larger models while requiring less time.

- **Effectiveness on policy model training:** Additional experiments requested by Reviewer ZALW show that 8B-model trained with P-GenRM achieves statistically significant gains in personalized alignment quality over 70B models.


**Summary of Reviewer Comments and Our Responses**

The reviewers highlighted several positive aspects of our work:

- Reviewer Ehiq: “a novel and well-argued design”, "motivation is clear and important"
- Reviewer qy8U:  “address an important problem of user personalization by providing the full pipeline”
- Reviewer ZALW: found P-GenRM’s performance promising with clear illustrations

All reviewers thought our experiments comprehensive.

The reviewers’ concerns centered on four main points, each of which we addressed in detail:

1. **Inference cost of P-GenRM**：We compared inference time and performance across open- and closed-source models. P-GenRM achieves higher accuracy (*75.92 vs. 72.05*) with lower latency than baselines (00:14:16 vs. at least 00:24:10). The proposed Test-time User-based Scaling further boosts performance by *3.24%*, with only a modest increase in inference cost.

2. **Additional evidence on the effectiveness of P-GenRM**：In response to the reviewers’ requests, we provided:

 $\quad \ $ (a) An analysis showing that P-GenRM requires as few as *3* historical samples per user to perform reliably;

 $\quad \ $ (b) Supplementary results with error bars to  confirm the soundness of each design choice and the statistical significance of our improvements over prior work.

$\quad \ $  (c) An evaluation across user groups of varying sizes using the *macro-accuracy* metric, demonstrating that P-GenRM remains robust even on minority groups.

3. **P-GenRM for policy model's training:**  We showed that 8B policy models trained with P-GenRM under DPO/GRPO settings outperform 70B-sized models, with statistically significant improvements.

4. **Clarification of writing issues:**  We provided further explanations and revised the paper to clarify the I/O setup, the intuition behind the three-stage training, the necessity of each component, a limitation analysis, and a clearer articulation of our contributions.

For your reference, the relation between the reviewers’ concerns, the four points above, and our corresponding revisions is outlined below:

|  | Addressed concerns  | Revisions in the manuscript  |
| :-- | :-- | -- |
| Point 1 $\quad$| Reviewer Ehiq: W2, Q2 a)  **\|**  Reviewer qy8U: W2  | Section Experiment, Appendix A.10, Table 11 |
| Point 2 $\quad$| Reviewer Ehiq: Q2 b)  **\|**  Reviewer ZALW: W1, W4   | Appendix A.12, Table 12 **\|**  Table 2/3/4  **\|**  Section Experiments, Appendix A.6, Figure 5, Table 9/10 |
| Point 3 $\quad$| Reviewer ZALW: W3, Q1, Q2  | Section 5.5, Appendix A.14, Table 14/15  |
| Point 4 $\quad$| Reviewer Ehiq: W1, Q1 **\|**  Reviewer qy8U: W1, Q1 **\|**  Reviewer ZALW: W2  $\quad$ | Section Methodology  **\|** Appendix A.15  |

Reviewer Ehiq found our clarifications reasonable and *increased the confidence to 3* while maintaining the score at 6.

Reviewer qy8U considered the concerns resolved and raised the score **from 4 to 6**.

Reviewer ZALW engaged in two thorough rounds of discussion and, after our further clarification on P-GenRM's benefits for  policy model  training, raised the score **from 4 to 6** in line with his previous statement to do so if his concerns were resolved.

**Finally, we sincerely thank you for your valuable time and careful consideration. Your dedication   to the research community is deeply valued**

---

### Meta-Review · Area_Chair_5c12 · 2026-01-06

**Summary:**

With the final scores converging to 6/6/6, I’m inclined to view this as a case where the rebuttal did real work rather than cosmetic repair—and that matters. Reviewers were already aligned on the fundamentals: Ehiq called the motivation “clear and important” and the design “novel and well-argued,” qy8U emphasized the value of a complete personalization pipeline, and ZALW found the headline results promising but initially unconvinced on rigor. What tipped the balance is that the authors systematically closed the open loops: they added concrete latency measurements and a credible systems explanation for why test-time scaling remains practical; they clarified the method’s I/O and the rationale for each training stage; and, crucially, they responded to statistical and downstream-utility concerns with new experiments—error bars, macro-accuracy to rule out majority-persona collapse, and policy-training results under DPO/GRPO with confidence intervals. ZALW’s explicit score increase after these additions is the clearest signal that the remaining doubts were resolved. This isn’t a flashy result, but it’s a careful one whose claims are now proportionate to the evidence, which is enough to justify acceptance.

**Reviewer Concerns:**

See above.

**Reviewer Scores:**

See above.

---

### Decision · Program_Chairs · 2026-01-26

Accept (Oral)